# Consistency and Unified Semantic Regularization for Generalized Category Discovery

## Abstract

Generalized Category Discovery (GCD) aims to leverage labeled data to learn clustering-friendly representations for unlabeled data. Among existing approaches, self-supervised contrastive learning (CL) is the most widely adopted, typically optimizing two objectives: `consistency` and `uniformity`. However, we observe an inherent tension between these objectives—while uniformity encourages a uniform distribution across the feature space, it can conflict with the goal of learning class-discriminative representations. To address this, we propose a two-stage framework that disentangles feature learning from self-contrastive objectives to better capture category concepts and represent auxiliary unlabeled data. In the first stage, the model constructs visual representations anchored to known category prototypes while reinforcing semantic links between labeled classes. The second stage extends this representation space to discover novel categories using a consistency objective combined with specifically designed regularization. Moreover, we introduce a novel `Semantic Exploration Energy mechanism` to capture shared semantics across categories, thereby mitigating the information loss caused by prototype orthogonalization. The proposed framework—Consistency and Unified Semantic Regularization (`CURE`)—retains the consistency objective and enhances it with semantic energy regularization. Our CURE achieves state-of-the-art performance across multiple benchmarks and significantly alleviates performance imbalance between known and novel classes.

## 1 Introduction

While supervised learning has demonstrated outstanding performance across a wide range of tasks, its success often comes at the cost of substantial data annotation efforts. This limitation has sparked increasing interest in weakly supervised approaches (Chen et al., 2020; He et al., 2020), which seek to reduce dependence on labeled data without sacrificing effectiveness. Within this context, Generalized Category Discovery (GCD) (Vaze et al., 2022a) has emerged as a promising paradigm that aims to learn category-level semantics from labeled data and leverage them to represent and cluster unlabeled data in a more scalable and practical manner.

CL has emerged as the most popular strategy for learning representations usable in GCD tasks. Broadly, existing methods either refine CL objectives (Rastegar et al., 2025; Choi et al., 2024) or enhance parametric modeling through prototype-based approaches (Wang et al., 2024; Ma et al., 2025; Zhang et al., 2023). A recent theoretical analysis (Wang & Isola, 2020) decomposes CL (Gutmann & Hyvärinen, 2010) into two fundamental components: `consistency`, which encourages invariance across augmented views, and `uniformity`, which spreads features uniformly on the hypersphere. While consistency is beneficial for GCD by promoting stable view-invariant representations, the uniformity pressure may weaken the natural density structure required for reliable class discovery, thereby limiting clustering performance.

Beyond the effect of uniformity, CL also encounters structural challenges that are particularly pronounced in GCD. Since InfoNCE treats all other samples as negatives, CL inherently suffers from false negatives or class collisions, where semantically similar samples are inadvertently pushed apart Huang et al. (2023). This behavior directly conflicts with the objective of forming compact clusters from unlabeled data. Moreover, CL typically relies on extremely large batch sizes or cross-batch feature queues to stabilize optimization and prevent representation collapse. However, these re-

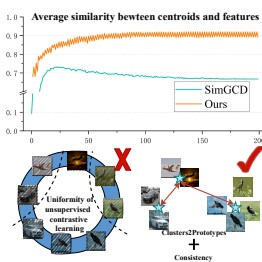
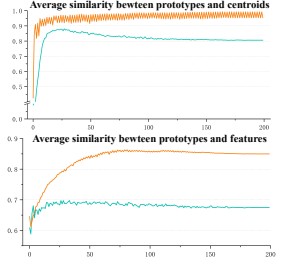

| Method | CUB | | | CIFAR100 | | |
|---|---|---|---|---|---|---|
| | All | Old | New | All | Old | New |
| w/o CL | 60.2 | 62.3 | 59.2 | 78.7 | 80.0 | 76.3 |
| w/o CL, resampling | 61.8 | 70.9 | 57.3 | 78.4 | 78.9 | 77.3 |
| CL of visual feature | 59.5 | 51.1 | 58.7 | 77.8 | 81.9 | 69.7 |
| SimGCD(original) | 60.3 | 65.6 | 57.7 | 80.1 | 81.2 | 77.8 |

Figure 1: CURE addresses the uniformity bias of contrastive learning by propagating semantics from labeled prototypes to unlabeled features. Through a Cluster-to-Prototypes mechanism, it aligns centroids with prototypes, improving both semantic transfer and clustering performance.

quirements pose significant challenges in GCD settings, where ViT-based models and limited GPU memory inherently restrict batch sizes.

Addressing the limitations imposed by uniformity remains a significant challenge in CL, as the outright removal of uniformity constraints often leads to representation collapse. To mitigate this issue, prior studies (Rastegar et al., 2025; Choi et al., 2024) have proposed relaxed formulations that balance the uniformity constraint with representation stability. Diverging from these approaches, we adopt a fundamentally different strategy: discarding the uniformity constraint and instead directly optimizing the semantic structure via a novel regularization method aimed at enhancing class-discriminative representations.

In the context of GCD, achieving class-discriminative representations hinges on effectively leveraging the semantic continuity between known and novel classes. However, commonly used one-hot supervision tends to overlook the intrinsic affinities between class prototypes, enforcing strict separability (Marino et al., 2017; Yang et al., 2022). This approach fragments the semantic space and hinders generalization to novel classes.

To alleviate the semantic discontinuity that arises in supervised objective, we reinterpret class prototypes as semantic attractors, rather than as orthogonal anchors typically emphasized in prior work. As illustrated in Figure 1, this perspective facilitates the structuring of the feature space, wherein soft alignment and energy-based regularization are employed to preserve and propagate semantic relationships across both labeled and unlabeled data.

Our framework unfolds in two stages. In the first stage, we utilize labeled data to construct a semantic topology among known class prototypes, where their relative positions reflect natural inter-class affinities. Instead of enforcing orthogonality, we encourage a soft relational encoding that captures nuanced semantic relationships. In the second stage, we guide the representation learning of unlabeled data by aligning them with the expanded prototype structure. By integrating structure-aware semantic exploration energy with consistency constraints, our method ensures semantic coherence across labeled and unlabeled domains. This design mitigates prototype fragmentation and presents a more flexible alternative to conventional contrastive uniformity assumptions.

Our main contributions are summarized as follows:

- We present **CURE**, the first GCD framework that eliminates the need for negative-sample-driven uniformity and learns representations purely from positive consistency and semantic clustering. Instead of uniformity, CURE builds representations solely through consistency regularization and semantic structuring.

- We propose a learnable semantic regularizer that preserves inter-class relations inferred from labeled data, encouraging coherent feature distributions and preventing prototype fragmentation. This is achieved via a C2P alignment that exchanges the class-level topology and instance-level representations learned by SEE, thereby enhancing semantic consistency and generalization to novel classes.

- We redesign the GCD training logic that prioritizes structured learning from labeled data via the proposed LGCS, which captures class-level semantics and instance representations by aligning features with prototypes in a shared space, preserving inter-class affinities with-

out orthogonality. This foundation enables accurate cluster inference on unlabeled data, facilitating robust C2P alignment and subsequent joint optimization over all data.

The proposed two-stage CURE framework represents a paradigm shift in GCD, introducing a structure-aware methodology rooted in consistency and semantic alignment rather than relying on CL. Extensive experiments conducted across seven public benchmarks demonstrate the efficacy of CURE, highlighting its ability to preserve discrimination among previously learned classes while significantly enhancing the discovery of novel ones.

## 2 RELATED WORK

### 2.1 CONTRASTIVE LEARNING

Contrastive learning distinguishes between similar and dissimilar data by constructing positive and negative sample pairs, thereby learning discriminative feature representations. SimCLR Chen et al. (2020) and SupCon Khosla et al. (2020) adopt the InfoNCE Gutmann & Hyvärinen (2010) loss, which treats all other samples as negatives. This design relies on a large number of negative examples to avoid feature collapse, and in practice often requires very large batch sizes, placing a heavy burden on memory and computation. MoCo He et al. (2020) addresses this issue by introducing a cross-batch feature queue maintained by a momentum encoder, allowing the model to access a rich set of negatives even under relatively small batch sizes.

Although contrastive learning performs well in unsupervised learning, it has inherent limitations in clustering and category discovery tasks. InfoNCE Gutmann & Hyvärinen (2010) induces false negatives/class collision issues, where semantically similar or even identical samples are erroneously pushed apart during training, thereby disrupting cluster structures. This behavior breaks the underlying cluster structure and degrades the quality of the learned representation. SCE Denize et al. (2023) and ProPos Huang et al. (2023) have highlighted class collision as a key factor behind the poor clustering behavior of contrastive objectives. In GCD settings, this issue becomes even more severe, which makes it particularly important to explore new paradigms that no longer rely on standard contrastive learning.

### 2.2 DEEP CLUSTERING

Deep clustering (Caron et al., 2018; 2019) aims to jointly learn representations and clustering assignments under fully unsupervised settings. Unlike traditional clustering methods (MacQueen, 1967; Law et al., 2017), which operate directly in the original space, deep clustering leverages neural networks to project high-dimensional data into a latent space that is more amenable to clustering.

A typical deep clustering framework consists of two key components: representation learning (Krizhevsky et al., 2012; He et al., 2016) and clustering (MacQueen, 1967). Reconstruction-based methods (Zhou et al., 2018; Mukherjee et al., 2019; Yang et al., 2023) are widely used to keep the input information while adding constraints to the latent space. For example, Deep Embedded Clustering (Xie et al., 2016) updates cluster assignments by minimizing the Kullback–Leibler (KL) divergence between the soft assignments and a target distribution. DeepCluster (Caron et al., 2018) alternates between clustering and supervised training with pseudo labels, so that both the features and the clusters are improved step by step.

Recent works further add contrastive learning (Ji et al., 2019), multi-view consistency (Van Gansbeke et al., 2020; Huang et al., 2020), and optimal-transport-based pseudo labeling (Asano et al., 2020) to improve cluster compactness and semantic separation. However, because they do not use any human-defined semantic signals, purely unsupervised models often fail to produce clusters that are easy to interpret or align well with human concepts. semantically meaningful clusters.

### 2.3 CATEGORY DISCOVERY

Category discovery transfers knowledge from human-labeled data to unlabeled data. In early work on New Category Discovery (NCD) (Han et al., 2022), the problem was formulated as pairwise relation predicted task, focusing on distinguishing whether a pair of samples belong to the same class (Hsu et al., 2019; Hsu & Kira, 2016; Zhao & Han, 2021).

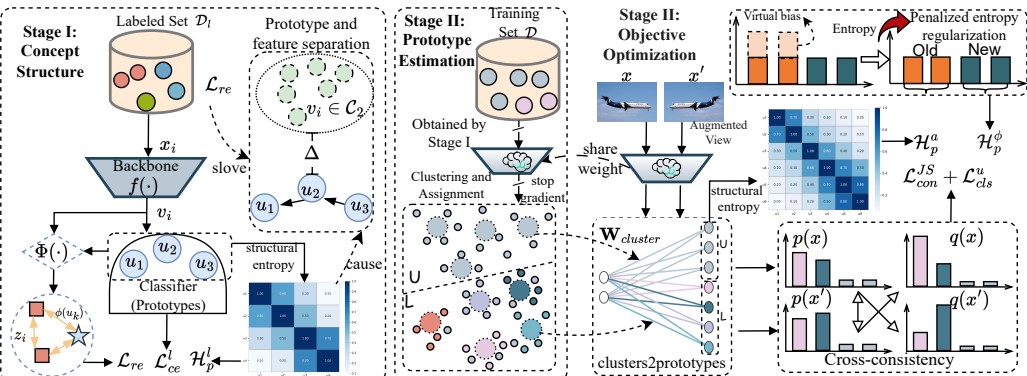

Figure 2: Overview of the proposed CURE approach. The training process is divided into two stages. **Stage I** initializes the semantic space by aligning labeled features with class prototypes while regularizing the structure using energy-based entropy. **Stage II** consists of two submodules: (1). prototype estimation via clustering, and (2). objective optimization that incorporates a virtual bias-aware entropy regularization and a JS-based consistency constraint. The two-stage design allows the model to gradually explore the semantic structure and mitigate local collapse.

A paradigm shift was introduced by UNO (Fini et al., 2021), which unified classification and clustering by training on pseudo labels generated via the SK algorithm (Asano et al., 2020; Cuturi, 2013). This marked the transition from pairwise modeling to more structured pseudo-supervision.

However, the NCD setting assumes that unlabeled data exclusively contains novel classes, which limits its applicability in real-world scenarios. Generalized Category Discovery (GCD) (Vaze et al., 2022a) addresses this limitation by relaxing the disjointness assumption and proposing a unified training strategy that combines supervised and unsupervised CL. GCD has since emerged as the dominant paradigm in this area, enabling joint optimization over known and unknown classes in a more realistic open-world setting (Pu et al., 2023; Choi et al., 2024; Rastegar et al., 2024).

Among recent methods, SimGCD (Wen et al., 2023) applies prototypical learning using pseudo-labels to enhance clusterability in the representation space, drawing inspiration from UNO's pseudo-supervised formulation. Building on this, parametric methods (Chiaroni et al., 2023; Vaze et al., 2023; Zhao et al., 2023; Liu & Han, 2025; Cao et al., 2024) have advanced the field by introducing more structured objectives, semantic prototypes, and regularization strategies that improve both representation discriminability and clustering robustness.

## 3 METHOD

### 3.1 PROBLEM DEFINITION

Let the training set $\mathcal{D}$ consist of a labeled subset $\mathcal{D}_l = \{(x_i, y_i)\}_{i=1}^{N_l}$ and an unlabeled subset $\mathcal{D}_u = \{x_j\}_{j=1}^{N_u}$. The labeled subset is drawn from a known class set $\mathcal{C}_l$, i.e., $y_i \in \mathcal{C}_l$. The unlabeled subset contains samples from both $\mathcal{C}_l$ and unknown classes $\mathcal{C}_u$, where $\mathcal{C}_l \cap \mathcal{C}_u = \emptyset$. The complete class space is defined as $\mathcal{C} = \mathcal{C}_l \cup \mathcal{C}_u$. Each instance $x_i$ is encoded by the backbone into the representation $v_i \in \mathbb{R}^d$. $\ell_2$ normalization is applied to $v_i$.

### 3.2 OVERVIEW

The core objective of the CURE is to mitigate the adverse effects of representation uniformity typically induced by CL. To achieve this, the framework adopts a two-stage design, as illustrated in Figure 2, which integrates a novel consistency regularization mechanism along with constraints on semantic structurality. Specifically, Stage I encodes semantic priors by aligning features with known prototypes under supervised guidance. Subsequently, Stage II progressively expands the prototype space to capture novel semantic structures through the combination of structure-aware clustering, bias-adjusted consistency regularization, and semantic alignment techniques.

### 3.3 SEMANTIC EXPLORATION ENERGY

In supervised learning settings (Chiaroni et al., 2023; Vaze et al., 2023; Zhao et al., 2023), cross-entropy loss is commonly employed to optimize the model under one-hot label supervision, encouraging the formation of feature-separable prototypes. But this type of training slowly pushes class prototypes toward being orthogonal. As a result, the prototype space can break into many separate parts, and the global semantic structure becomes weak. This problem is more serious in the GCD setting, where new classes need to stay close to known classes in the semantic space so that the model can align and recognize them.

To address this issue, we introduce a soft regularizer termed Semantic Exploration Energy (SEE). Specifically, let $\mathcal{P} = \{\mu_1, \mu_2, \ldots, \mu_M\}$ denote the set of $M$ prototypes. The pairwise cosine similarity is given by $S_{ij} = \cos(\mu_i, \mu_j)$, and the energy is defined as:

$$\mathcal{E} = \frac{1}{M-1} \sum_{i<j} (1 - S_{ij})^2 = \frac{1}{M-1} \|\mathbf{1}_M - S\|_F^2. \tag{1}$$

A large value of $\mathcal{E}$ means that prototypes have weak semantic links, while a smaller value lets them overlap in a soft way and share common concepts.

We then use a smooth and differentiable penalty to keep semantic cohesion:

$$\mathcal{R}_{\text{SE}} = \log\left(1 + \exp\left(\mathcal{E} - \epsilon\right)\right), \tag{2}$$

where $\epsilon$ controls how spread out the prototypes are allowed to be. When $\mathcal{E} > \epsilon$, this term adds a larger penalty, and the effect is similar to a soft ReLU. In this way, SEE does not force strict orthogonality. It instead gives flexible control and supports the discovery of new classes around existing semantics, even when the supervision is weak.

Theoretically, the regularizer above encourages prototypes to lie on a manifold with low tension, so that the model can move smoothly from known regions to unknown ones in the semantic space. But SEE acts only on prototype–prototype relations. It makes class centers semantically connected, yet it does not directly make sure that this global structure appears in the feature space. In practice, the softmax classifier is a local smooth version of the argmax rule, and it may not pass prototype relations down to feature representations. This gap leads to the next part of our framework, which we call label-guided concept structure, whose goal is to connect global prototype exploration with local feature alignment.

### 3.4 LABEL-GUIDED CONCEPT STRUCTURE

Based on the semantic links given by SEE, we next ensure that the relationships between prototypes are transferred into the feature space. The cross-entropy loss on labeled data does push the model to learn discriminative features, and it updates the prototype set $\mathcal{P}$ by minimizing the empirical classification risk:

$$\mathcal{P}^* = \arg\min_{\mathcal{P}} \ \mathbb{E}_{(\mathbf{x},y)\sim\mathcal{D}_l} \left[ -\log \frac{\exp(\mathbf{p}_y^\top f(\mathbf{x}))}{\sum_j \exp(\mathbf{p}_j^\top f(\mathbf{x}))} \right]. \tag{3}$$

This objective drives each prototype $\mathbf{p}_j$ to increase its dot-product similarity with features from its own class. Once converged, the learned classifier approximates the optimal decision function by:

$$\arg\max_j \ \mathbf{p}_j^\top f(\mathbf{x}) = \arg\max_j \ \log p(y = j|\mathbf{x}), \tag{4}$$

where $p(y = j|\mathbf{x})$ is the posterior probability from the softmax layer. In short, the classifier gives a smooth version of the argmax boundary and behaves almost linearly near class changes.

This process works well for local discrimination, but it does not by itself spread the semantic relations found by SEE to all features. To close this gap, we design a prototype–feature alignment term in a shared semantic space. Let $\Phi(\cdot)$ be an MLP projection applied to both prototypes and sample features. For a sample feature $v_i$, we get a projected feature $z_i = \Phi(v_i)$. The alignment loss has two parts: self-to-prototype alignment and class-aware alignment. For each labeled feature $z_i$, we push

it toward its own semantic prototype and also toward other features $z_j$ in the same class:

$$\mathcal{L}_{\text{align}}^{\text{self}} = \frac{-1}{|B|} \sum_{i=1}^{|B|} \log \left( \frac{\exp \left( z_i^\top \Phi(u_{y_i})/\tau \right)}{Z_i} \right), \quad \mathcal{L}_{\text{align}}^{\text{class}} = \frac{-1}{|B||\mathcal{N}_i|} \sum_{i=1}^{|B|} \sum_{j \in \mathcal{N}_i} \log \left( \frac{\exp \left( z_i^\top z_j/\tau \right)}{Z_i} \right),$$
(5)

where $u_{y_i}$ is the prototype of the ground-truth class, $\tau$ is a temperature, and $\mathcal{N}i$ is the set of samples in the batch that share the same label as sample $i$. The normalizer sums over both prototypes and nearby features:

$$Z_i = \sum_{j=1}^{|B|} \exp \left( z_i^\top z_j/\tau \right) + \exp \left( z_i^\top \Phi(u_{y_i})/\tau \right).$$
(6)

The final alignment objective is defined as a weighted combination:

$$\mathcal{L}_{\text{align}} = \alpha \mathcal{L}_{\text{align}}^{\text{self}} + (1 - \alpha) \mathcal{L}_{\text{align}}^{\text{class}},$$
(7)

which encourages prototypes to serve as global semantic anchors aligned with both sample-level features and class-level distributions.

The Stage-I training loss thus becomes:

$$\mathcal{L}_{\text{stage-I}} = \mathcal{L}_{\text{CE}} + \mathcal{L}_{\text{align}} + \mathcal{R}_{\text{SE}}^l,$$
(8)

where $\mathcal{R}_{\text{SE}}^l$ denotes the SEE regularizer applied on labeled prototypes. Together, SEE and LGCS form a two-level semantic regularization scheme: SEE preserves global prototype connectivity, while LGCS ensures that such connectivity is faithfully propagated to the feature space, establishing a coherent and unified semantic structure for category discovery.

### 3.5 STRUCTURE-GUIDED SEMANTIC EXPANSION

After supervised pretraining on the labeled dataset $\mathcal{D}_l$, the model has learned the key semantics of the known class space $\mathcal{C}_l$. But due to the local nature of the cross-entropy loss, the prototypes focus more on being discriminative than on being representative, and the semantic space can still be fragmented. To improve this part, we design a structure-guided prototype reconstruction step. This step lets semantics move from labeled to unlabeled data by updating the prototype space with the concept structure built on the mixed dataset $\mathcal{D}$.

**Clusters to Prototypes.** Given frozen model parameters $\theta$, we extract features from the dataset $\mathcal{D}$, producing a feature set $\mathcal{F} = \{\mathbf{v}_i\}_{i=1}^{|\mathcal{D}|}$. We then perform hierarchical clustering - Ward linkage (Ward Jr, 1963) on $\mathcal{F}$ to obtain cluster assignments $\mathcal{G} = \{\mathcal{G}_1, ..., \mathcal{G}_{|\mathcal{C}|}\}$. Each cluster $\mu_k = \frac{1}{|\mathcal{G}_k|} \sum_{\mathbf{v}_i \in \mathcal{G}_k} \mathbf{v}_i$, serves as a prototype initialization.

To semantically guide these clusters, we use features from $\mathcal{D}_l$ as anchors and apply Hungarian matching (Kuhn, 1955) to align a subset of clusters to known classes $\mathcal{C}_l$. The remaining clusters are retained as novel class candidates $\mathcal{C}_u$, resulting in the full set of prototypes $\mathcal{P} = \{\mu_1, ..., \mu_{|\mathcal{C}|}\}$.

This strategy guarantees structural coherence in the feature space while preserving semantic consistency with known categories. The use of Ward linkage minimizes intra-cluster variance and approximates a local optimum under Gaussian assumptions, making the resulting centroids a principled initialization for semantic optimization.

**Semantic Consistency Optimization** Next, we optimize the prototypes $\mathcal{P} = \{\mu_1, \ldots, \mu_{|\mathcal{C}|}\}$ based on the dataset $\mathcal{D}$ by using a joint loss composed of consistency constraints and multiple regularization. Specifically, for each sample $x_i \in \mathcal{D}$, we generate two augmented views and compute the corresponding predictions $p_i^{(v_1)}, p_i^{(v_2)}$. The semantic consistency can then be achieved by minimizing the symmetrical divergence:

$$\mathcal{L}_{\text{KL}} = \frac{1}{2} \text{KL}(p_i^{(v_1)} \| p_i^{(v_2)}) + \frac{1}{2} \text{KL}(p_i^{(v_2)} \| p_i^{(v_1)}).$$
(9)

This loss encourages predictions to be invariant under view transformations and reduces the risk of prototype overfitting to specific augmentations, serving as a soft clustering signal that aligns with the cluster assumption. On the other hand, a cross-entropy loss $\mathcal{L}_{CLS}^l$ is used for the labeled set, enabling consistency between model predictions and ground-truth labels.

**Logit-aware Self-Distillation.** Inspired by SimGCD (Wen et al., 2023), we also introduce a self-distillation objective for sample $x_i \in \mathcal{D}_u$ between predictions at different temperature scales, where the low-temperature prediction $q_i^{(v_1)}$ serves as the teacher to supervise the high-temperature prediction $p_i^{(v_2)}$. Furthermore, a compatible logit adjustment (Chen et al., 2022) is adopted for self-distillation loss. This adjustment introduces class-frequency-aware priors to reduce the bias toward known classes, as previously discussed in (Chen et al., 2022), and enhances compatibility between supervised and unsupervised distributions. Concretely, we compute the mean frequency $\bar{f}_k$ of known classes $f_i$ based on labeled data, and use it as a soft bias added to the predicted probability of unseen classes. The adjusted prediction distribution $q^{\text{adj}}$ is defined as:

$$\tilde{p}_{i,k} = \begin{cases} p_{i,k} + f_k, & \text{if } k \in \mathcal{C}_l \\ p_{i,k} + \bar{f}_k, & \text{if } k \in \mathcal{C}_u \end{cases} \tag{10}$$

$$\mathcal{L}_{CLS}^u = -\frac{1}{2} \sum_{k=1}^{\mathcal{C}} \left[ q_i^{(v_1)} \log \tilde{p}_i^{(v_2)} + q_i^{(v_2)} \log \tilde{p}_i^{(v_1)} \right]. \tag{11}$$

The final function of the consistency is $\mathcal{L}_{\text{cons}} = \mathcal{L}_{\text{JS}} + \mathcal{L}_{CLS}$, and $\mathcal{L}_{CLS} = \frac{1}{2}\mathcal{L}_{CLS}^l + \frac{1}{2}L_{CLS}^u$.

**Virtual Sampling and Entropy Regularization** This approach circumvents the sampling-induced bias and instability often introduced by re-weighting or re-sampling. Instead, it directly adjusts the energy landscape of the prediction logits, encouraging uniform exploration of novel classes while maintaining predictive confidence on known categories.

Instead of altering the sampling distribution, we modify the input logits to softmax by applying a structural penalty to all known classes $\mathcal{C}_l$. Let $\mathbf{o}_i \in \mathbb{R}^{|\mathcal{C}|}$ denote the raw logits for sample $\mathbf{x}_i$. We define the penalized logits as:

$$\tilde{\mathbf{o}}_i = [t \cdot o_{i,1}, \ldots, t \cdot o_{i,\mathbf{C}_l}, o_{i,\mathbf{C}_l+1}, \ldots, o_{i,|\mathcal{C}|}], \tag{12}$$

where $t$ is a fixed scaling coefficient that adjusts the logits of known classes. The softmax is then applied to $\tilde{\mathbf{o}}_i$, yielding the adjusted probability distribution $\tilde{p}_i = \text{softmax}(\tilde{\mathbf{o}}_i)$. The adjusted entropy regularization $\mathcal{H}(\tilde{\mathbf{o}}_i)$ is then calculated in the manner of (Wen et al., 2023).

**Semantic Energy Regularization.** Unlike the $\mathcal{R}_{\text{SE}}^l$ in the first-stage, which was applied to the supervised prototype set $\mathcal{P}^l$, we now compute a new semantic regularizer over the full set of prototypes $\mathcal{P}$, where each prototype corresponds to the centroid of a discovered cluster $\mathcal{G}_k$.

While the first-stage regularizer $\mathcal{R}_{\text{SE}}^l$ aims to preserve a semantically connected prototype space, the second-stage energy regularization $\mathcal{R}_{\text{SE}}$ focuses on contracting the distances between prototypes. This contraction enables unlabeled instances, which are initially assigned to different clusters via a nearest-centroid rule, to escape strict prototype binding under the influence of $\mathcal{L}_{cons}$.

The overall objective is formulated as a composite loss, and defined as:

$$\mathcal{L} = \mathcal{L}_{\text{cons}} + \mathcal{H}(\tilde{\mathbf{o}}) + \lambda \cdot \mathcal{R}_{\text{SE}}, \tag{13}$$

where $\lambda$ is a hyperparameter to balance the influence of the regularization term during optimization.

# 4 EXPERIMENTS

## 4.1 EXPERIMENTAL SETUP

**Datasets and Evaluation Protocol.** The performance of the proposed CURE framework is thoroughly evaluated on six publicly available benchmark datasets (CIFAR-10/100 (Krizhevsky et al., 2009), ImageNet-100 (Russakovsky et al., 2015), CUB-200 (Reed et al., 2016), Stanford-Cars (Krause et al., 2013), and Herbarium19 (Tan et al., 2019)). To ensure a fair and consistent comparison with prior works, we follow the widely adopted dataset partitioning protocol (Vaze et al., 2022a), which divides each dataset into a labeled and an unlabeled subset. All reported clustering accuracy (ACC) are computed on the standard evaluation splits under this protocol.

**Implementation Details.** we adopt the Vision Transformer (ViT-B/16) (Dosovitskiy et al., 2021) as backbone pretrained with the DINO (Caron et al., 2021) algorithm, and use the [CLS] token as

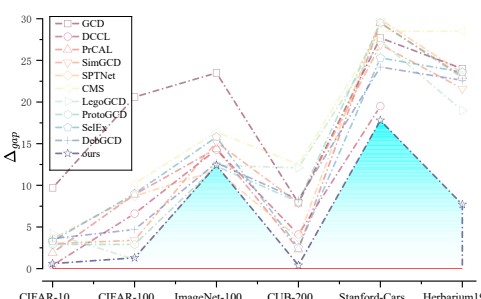 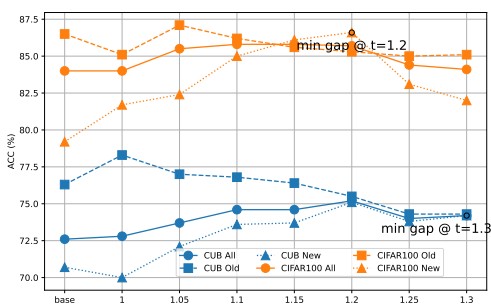

Figure 3: Absolute accuracy gap between Old and New Classes. Our method consistently achieves the smallest gaps.

Figure 4: Effect of virtual sampling multiplier $t$ on ACC and the gap between Old and New Classes.

the visual representation. The training process is divided into two stages. In the first stage, we freeze the first $n$ transformer blocks (responsible for extracting low-level features) and fine-tune the subsequent layers using the labeled data. In the second stage, we unfreeze and jointly fine-tune the last block using both labeled and unlabeled data (the ablation experiment is shown in the appendix).

For hyperparameter settings, the weighting coefficient $\alpha$ is set to 0.35, and the virtual sampling rate $t$ is fixed at 1.2. The regularization weight $\lambda$ is set to 2 for coarse-grained datasets and 20 for fine-grained datasets, to account for their differing semantic granularity (more detailed analysis can be found in the appendix.).

## 4.2 COMPARISON WITH STATE-OF-THE-ART

As shown in Table 1, our method achieves overall leading precision across a variety of datasets. More importantly, our approach exhibits significantly more balanced performance between old and novel classes, as shown in Figure 3, which is a crucial indicator of generalization in GCD.

Specifically, we achieve 85.7% and 87.0% on All classes on CIFAR-100 and ImageNet-100, respectively. In both cases, the performance gap between old and novel class is notably small, As shown in Figure 3, demonstrating the ability of CURE to learn unbiased representations. In contrast, many state-of-art methods suffer from skewed predictions. These methods, like SimGCD, SPTNet, LegoGCD and DebGCD, achieves a strong result on old classes, but shows a marked drop on novel classes, indicating a clear imbalance.

This balanced accuracy is especially critical for fine-grained datasets, where class boundaries are more subtle. On CUB-200 and Stanford-Cars, our method yields minimal accuracy gaps between old and new classes, suggesting effective semantic modeling across closely related categories. Furthermore, on the highly challenging Herbarium19 dataset, our method not only achieves the highest overall accuracy but also delivers the smallest performance gap among all methods.

## 4.3 ABLATION STUDY

To better understand the role of each component in our framework, we isolate and disable each of these components individually to evaluate its independent contribution on CUB-200 and CIFAR-100. Experimental results are reported to provide a clear view of their effects. Table 2 shows the results obtained in Experiment 1 by replacing the random initialization of prototypes with C2P combined with standard supervised learning pre-training.

**The effectiveness of supervised fine-tuning.** The ablation studies clearly demonstrate the feasibility of incorporating strong supervision during the stage I. Our proposed prototype-to-feature alignment loss, combined with semantic regularization, effectively mitigates the tendency of standard cross-entropy (with one-hot targets) to produce orthogonalized class representations. This orthogonalization often results in the loss of shared semantics across categories. In contrast, our approach preserves inter-class semantic information and facilitates the transition from known to novel classes. This advantage is particularly evident from the comparisons between Experiments 1 and 2, and Experiments 7 and 8 in Table 2.

Table 1: Performance comparison across six benchmark datasets. Our method achieves consistently strong results on both coarse- and fine-grained data. **Note:** The best results are **bolded** and the second-best results are underlined.

| Methods | CIFAR-10 | | | CIFAR-100 | | | ImageNet-100 | | | CUB-200 | | | Stanford-Cars | | | Herbarium19 | | |
|---|---|---|---|---|---|---|---|---|---|---|---|---|---|---|---|---|---|---|
| | All | Old | New | All | Old | New | All | Old | New | All | Old | New | All | Old | New | All | Old | New |
| GCD | 91.5 | 97.9 | 88.2 | 70.8 | 77.6 | 57.0 | 74.1 | 89.8 | 66.3 | 51.3 | 56.6 | 48.7 | 39.0 | 57.6 | 29.9 | 35.4 | 51.0 | 27.0 |
| DCCL | 96.3 | 96.5 | 96.9 | 75.3 | 76.8 | 70.2 | 80.5 | 90.5 | 76.2 | 63.5 | 60.8 | 64.9 | 43.1 | 55.7 | 36.2 | - | - | - |
| PrCAL | **97.9** | 96.6 | 98.5 | 81.2 | 84.2 | 75.3 | 83.1 | 92.7 | 78.3 | 62.9 | 64.4 | 62.1 | 50.2 | 70.1 | 40.6 | 37.0 | 52.0 | 28.9 |
| SimGCD | 97.1 | 95.1 | **98.1** | 80.1 | 81.2 | 77.8 | 83.0 | 93.1 | 77.9 | 60.3 | 65.6 | 57.7 | 53.8 | 71.9 | 45.0 | 44.0 | 58.0 | 36.4 |
| SPTNet | 97.3 | 95.0 | **98.6** | 81.3 | 84.3 | 75.6 | 85.4 | 93.2 | 81.4 | 65.8 | 68.8 | 65.1 | 59.0 | 79.2 | 49.3 | 43.4 | 58.7 | 35.2 |
| CMS | - | - | - | 82.3 | **85.7** | 75.5 | 84.7 | **95.6** | 79.2 | 68.2 | **76.5** | 64.0 | 56.9 | 76.1 | 47.6 | 36.4 | 54.9 | 26.4 |
| LegoGCD | 97.1 | 94.3 | 98.5 | 81.8 | 81.4 | **82.5** | 86.3 | 94.5 | 82.1 | 63.8 | 71.9 | 59.8 | 57.3 | 75.7 | 48.4 | 45.1 | 57.4 | 38.4 |
| ProtoGCD | 97.3 | 95.3 | 98.2 | 81.9 | 82.9 | 80.0 | 84.0 | 92.2 | 79.9 | 63.2 | 68.5 | 60.5 | 53.8 | 73.7 | 44.2 | 44.5 | **59.4** | 36.5 |
| SelEx | 95.9 | **98.1** | 94.8 | 82.3 | 85.3 | 76.3 | 83.1 | 93.6 | 77.8 | 73.6 | 75.3 | 72.8 | 58.5 | 75.6 | 50.3 | 39.6 | 54.9 | 31.3 |
| DebGCD | 97.2 | 94.8 | 98.4 | 83.0 | 84.6 | 79.9 | 85.9 | 94.3 | 81.6 | 66.3 | 71.8 | 63.5 | 65.3 | **81.6** | 57.4 | 44.7 | **59.4** | 36.8 |
| ours | 97.5 | 97.9 | 97.3 | **85.7** | 85.3 | **86.6** | **87.0** | 95.3 | **82.9** | **75.2** | 75.5 | **75.1** | **68.7** | 80.8 | **63.0** | **48.1** | 53.1 | **45.4** |

Table 2: Ablation study of different components on CUB-200 and CIFAR-100. *FT* indicates supervised fine-tuning designed in the first stage.

| Component | $FT$ | $\mathcal{R}_{SE}$ | $\mathcal{L}_{SKL}$ | CUB-200 | | | CIFAR-100 | | |
|---|---|---|---|---|---|---|---|---|---|
| | | | | All | Old | New | All | Old | New |
| 1 | | | | 70.4 | 72.9 | 69.2 | 81.2 | 85.5 | 72.5 |
| 2 | ✓ | | | 72.1 | 75.1 | 70.7 | 83.5 | 85.2 | 80.0 |
| 3 | | ✓ | | 73.7 | 77.7 | 71.7 | 84.0 | 85.3 | 81.5 |
| 4 | | | ✓ | 72.8 | 74.7 | 71.9 | 85.0 | 6.4 | 82.3 |
| 5 | ✓ | ✓ | | 73.0 | 75.2 | 72.0 | 84.6 | 86.3 | 81.3 |
| 6 | ✓ | | ✓ | 74.0 | 73.9 | 74.0 | 85.6 | 86.5 | 84.1 |
| 7 | | ✓ | ✓ | 72.6 | 75.9 | 70.9 | 85.3 | 86.5 | 83.1 |
| 8 | ✓ | ✓ | ✓ | 75.2 | 76.2 | 74.7 | 85.2 | 86.4 | 82.9 |

**Effectiveness of $\mathcal{R}_{SE}$.** we conduct several experiments to analyze the ability of semantic regularization to extend from old classes to new classes. As shown in Table 2, semantic regularization in the second stage significantly improves performance, even in different module combinations. It likely facilitates a better global structure or class-aware separation, which is critical for semantic generalization.

**Effectiveness of symmetrical divergence.** To assess the role of the JS divergence, we compare the performance with and without this component. Removing the JS loss leads to a noticeable drop in accuracy, especially on new classes. This confirms that JS divergence plays a key role in maintaining stability between different views, particularly under distributional shifts. The symmetric nature of JS divergence encourages alignment between different views while avoiding mode collapse.

**Effectiveness of different clustering methods.** To obtain cluster assignments, we further evaluated several clustering algorithms, including K-means, Gaussian mixture models (GMM), spectral clustering and hierarchical clustering, as shown in Table 3. The results indicate that K-means and GMM significantly underperform compared to spectral and hierarchical clustering. We attribute this to the limited capacity of K-means and GMM in modeling complex distributions, which restricts their ability to capture the underlying structure in high-dimensional semantic spaces.

## 4.4 HYPER-PARAMETER ANALYSIS

To investigate the effect of the virtual sampling multiplier $t$ and the influence of the semantic energy threshold $\epsilon$, we conduct experiments on both CUB-200 and CIFAR-100, adjusting the value of $t$ from 1.0 to 1.3 and the value of $\epsilon$ from 1.0 to 0.8. As shown in Figure 4 and Table 4.

**Effect of the virtual sampling multiplier $t$.** we observe that the accuracy gap between old and new classes consistently narrows as $t$ increases. For CUB-200 dataset, the gap decreases from 5.6% at the base setting to just 0.1% when $t = 1.3$. A similar trend is observed on CIFAR-100, where the

Table 3: Ablation study of CURE with different Clustering methods to construct centroids.

| Clustering | CUB-200 | | | CIFAR-100 | | |
|---|---|---|---|---|---|---|
| | All | Old | New | All | Old | New |
| Kmeans | 73.4 | 74.2 | 72.9 | 84.9 | 85.2 | 84.5 |
| GMM | 73.8 | 75.1 | 73.1 | 84.6 | 85.1 | 83.7 |
| Spectral | 76.7 | 77.3 | 76.3 | 85.6 | 85.0 | 86.9 |
| Hierarchy | 75.2 | 75.5 | 75.1 | 85.7 | 85.3 | 86.6 |

Table 4: Performance under different values of the semantic energy threshold $\epsilon$.

| $\epsilon$ | CUB-200 | | | CIFAR-100 | | |
|---|---|---|---|---|---|---|
| | All | Old | New | All | Old | New |
| 1.00 | 73.6 | 74.4 | 73.2 | 85.3 | 85.2 | 85.5 |
| 0.95 | 74.6 | 75.1 | 74.4 | 85.8 | 85.5 | 86.3 |
| 0.90 | 74.7 | 74.9 | 74.6 | 85.5 | 85.2 | 86.2 |
| 0.85 | 74.5 | 75.9 | 73.8 | 85.5 | 85.0 | 86.6 |
| 0.80 | 73.2 | 75.3 | 72.2 | 84.1 | 85.1 | 82.0 |

Table 5: Comparison of training time, inference time, and overall accuracy on ImageNet-100 and SSB. All times are measured in seconds.

| Method | ImageNet-100 | | | SSB | | |
|---|---|---|---|---|---|---|
| | Acc.(All) | Train(s) | Infer(s) | Acc.(All) | Train(s) | Infer(s) |
| GCD | 74.1 | 803 | 2289 | 51.3 | 58 | 552 |
| SimGCD | 83.0 | 847 | 591 | 56.1 | 64 | 17 |
| PromptCAL | 83.1 | 1817 | 893 | 55.1 | 492 | 103 |
| SPTNet (w/ SimGCD initialization) | 85.4 | 483* | 601 | 61.4 | 32* | 17 |
| CURE (C2P with $k$-epoch alignment) | - | 50 | - | - | 44 | - |
| CURE Stage-1 (LGCS on labeled data) | 83.2 | 152 | - | 58.0 | 13 | - |
| CURE Stage-2 (full training) | 87.0 | 375 | 315 | 68.5 | 49 | 17 |

*SPTNet requires a pre-trained SimGCD model, so its actual total training time is: SimGCD (847s) + SPTNet (483s) = 1330s on ImageNet-100; similarly, 64s + 32s = 96s on SSB.

gap is reduced from 7.3% to 1.3% at $t = 1.2$. In order to balance ACC accuracy, $t$ is set to 1.2 as the optimal value.

**Effect of the Semantic Energy Threshold** $\epsilon$. As shown in Table 4, the semantic energy threshold $\epsilon$ controls the degree of orthogonality between prototypes. A higher threshold allows more semantic overlap between classes but risks diluting the discriminative power by collapsing shared semantics. Conversely, a lower threshold enforces stronger separation, which may overly suppress inter-class variability and result in non-separable feature spaces. Empirical results demonstrate that a wide range of thresholds within $[0.85, 0.95]$ yields consistently strong performance on both datasets. This indicates that the proposed semantic regularization effectively as a soft constraint, exhibiting robustness to the exact choice of $\epsilon$.

### 4.5 TIME ANALYSIS ON DIFFERENT METHODS

To check the running time of different GCD methods, we look at training time, inference time, and accuracy on ImageNet-100 and SSB. Table 5 shows the results. CURE keeps costs low even with its two-stage setup. Stage 1 runs on labeled data only and takes little time. It needs 152 seconds on ImageNet-100 and 13 seconds on SSB. Stage 2 uses 375 seconds and 49 seconds on those datasets. These times match or beat most other methods. The total time stays low, but accuracy reaches 87.0% on ImageNet-100 and 68% on SSB—better than the rest. Some methods cost more. PromptCAL takes 1817 seconds to train on ImageNet-100. That is over twice as long as CURE, and its accuracy is lower. GCD and SimGCD also run longer overall. Their inference times are high, like 2289 seconds for GCD. SPTNet trains in 483 seconds by itself. But it starts with a full SimGCD model. The real total adds up to 847 seconds plus 483 seconds, or 1330 seconds on ImageNet-100. This makes SPTNet one of the slowest. Our C2P step updates every few rounds and adds just 50 seconds or 44 seconds. It shows the alignment costs little. CURE gives the best mix of speed and accuracy. It skips heavy pre-training. It avoids big costs from prompting or extra stages, like in SPTNet. The two-stage plan keeps everything low-cost.

## 5 CONCLUSION

In this work, we explore the feasibility of leveraging supervised signals during pretraining to decouple GCD from the CL, thereby mitigating the adverse effect of uniformity on novel category discovery. Our core innovation lies in uncovering the semantic dependencies among prototypes, which are crucial for guiding the representation learning of unlabeled samples. To preserve and utilize these semantic relations, we introduce a semantic exploration energy mechanism that prevents their degradation during training. Furthermore, the proposed Clusters-to-Prototypes (C2P) mechanism enables the effective transfer of structural semantics from the prototype space to the feature space. Extensive experiments across multiple datasets confirm the effectiveness of our approach.

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

# A THEORETICAL ANALYSIS OF SEMANTIC EXPLORATION ENERGY

## A.1 SPECTRAL ANALYSIS OF SEMANTIC EXPLORATION ENERGY

To justify the design of Semantic Exploration Energy (SEE), we analyze its effect on the spectral properties of the prototype similarity graph.

Let $\mathcal{P} = \{\mu_1, \mu_2, \ldots, \mu_M\}$ denote $M$ class prototypes with unit norm, i.e., $\|\mu_i\| = 1$. Define the cosine similarity matrix $S \in \mathbb{R}^{M \times M}$ with $S_{ij} = \cos(\mu_i, \mu_j)$, and construct the degree matrix $D = \mathrm{diag}(S\mathbf{1})$. The unnormalized graph Laplacian is $L = D - S$.

The SEE energy is defined as:

$$\mathcal{E} = \frac{1}{M-1} \sum_{i<j} (1 - S_{ij})^2 = \frac{1}{M-1} \|\mathbf{1}_M - S\|_F^2. \tag{14}$$

This quantity measures the total squared deviation of pairwise similarities from perfect alignment ($S_{ij} = 1$). It penalizes overly orthogonal prototypes and encourages soft coupling.

**Proposition.** Minimizing $\mathcal{E}$ under the constraint $\|\mu_i\| = 1$ increases the algebraic connectivity $\lambda_2(L)$ of the semantic graph, i.e.,

$$\frac{\partial \mathcal{E}}{\partial \mu_i} < 0 \quad \Rightarrow \quad \frac{\partial \lambda_2(L)}{\partial \mu_i} > 0.$$

**Sketch of Proof.** First, note that $\mathcal{E}$ is minimized when the off-diagonal elements of $S$ are close to 1, i.e., prototypes are highly aligned.

Consider the spectral decomposition of $L = U\Lambda U^\top$, where $\Lambda = \mathrm{diag}(\lambda_1, \ldots, \lambda_M)$. The second smallest eigenvalue $\lambda_2(L)$, known as the Fiedler value, measures the graph's algebraic connectivity.

From matrix perturbation theory:

$$\frac{\partial \lambda_2}{\partial S_{ij}} = u_2(i)^2 + u_2(j)^2 - 2u_2(i)u_2(j),$$

where $u_2$ is the Fiedler eigenvector. Increasing $S_{ij}$ (i.e., aligning prototypes $\mu_i$ and $\mu_j$) directly increases $\lambda_2$, especially when $u_2(i)$ and $u_2(j)$ differ, i.e., when $\mu_i$ and $\mu_j$ lie on different components of the graph.

Since $\mathcal{E}$ penalizes $(1 - S_{ij})^2$, minimizing $\mathcal{E}$ increases many $S_{ij}$ values and therefore lifts $\lambda_2(L)$ globally.

**Implication.** As $\lambda_2(L)$ increases, the semantic prototype graph becomes more connected. According to Cheeger's inequality:

$$\frac{\lambda_2(L)}{2} \leq \Phi(G) \leq \sqrt{2\lambda_2(L)\Delta},$$

where $\Phi(G)$ is the Cheeger constant and $\Delta$ is the maximum degree. Thus, minimizing SEE improves both the spectral and combinatorial connectivity of the semantic structure, allowing novel classes to be inserted into a coherent, continuous topology.

# B ADDITIONAL DETAILS

## B.1 DETAILS OF BENCHMARK DATASETS

In addition to the six publicly available benchmark datasets discussed in the main text, this section further introduces the fine-grained Aircraft dataset (Maji et al., 2013), which is extensively adopted in Generalized Category Discovery (GCD) research. Comparative results on this dataset conclusively demonstrates the superiority of the proposed algorithm.

Table 6: Data distribution and partitioning protocol of coarse-grained benchmarks.

|  |  | CIFAR-10 | CIFAR-100 | ImageNet-100 |
|---|---|---|---|---|
| $\mathcal{D}_l$ | $|\mathcal{C}_l|$ | 5 | 80 | 50 |
|  | $|\mathcal{D}_l|$ | 12.5k | 20k | 31.9k |
| $\mathcal{D}_u$ | $|\mathcal{C}_u|$ | 10 | 100 | 100 |
|  | $|\mathcal{D}_u|$ | 37.5k | 30k | 95.3k |

Table 7: Data distribution and partitioning protocol of fine-grained benchmarks.

|  |  | CUB-200 | Stanford-Cars | Herbarium19 | Aircraft |
|---|---|---|---|---|---|
| $\mathcal{D}_l$ | $|\mathcal{C}_l|$ | 100 | 98 | 341 | 50 |
|  | $|\mathcal{D}_l|$ | 1.5k | 2.0k | 8.9k | 1.7k |
| $\mathcal{D}_u$ | $|\mathcal{C}_u|$ | 200 | 196 | 683 | 200 |
|  | $|\mathcal{D}_u|$ | 4.5k | 6.1k | 25.4k | 5.0k |

Table 6 and Table 7 summarize the distribution of labeled and unlabeled data in seven datasets, as well as the number of categories for novel classes and known classes. The data partitioning strictly adheres to the standard Generalized Category Discovery (GCD) protocol: For coarse-grained datasets (CIFAR-10 and ImageNet-100), the first 50% of classes are designated as known classes, while the latter 50% constitute novel classes. CIFAR-100 follows an alternative split where the initial 80% of classes are categorized as known, with the remaining 20% treated as novel. For fine-grained datasets, the data partitioning follows the Shift Semantic Benchmark (Vaze et al., 2022b) criteria. Crucially, the unlabeled dataset comprises all samples from novel classes combined with a randomly selected 50% subset of known class samples, whereas the labeled set exclusively consists of the remaining 50% of known class samples.

## B.2 DETAILS OF EVALUATION PROTOCOL

In the Generalized Category Discovery (GCD) task, the evaluation protocol must capture both classification performance on known classes and clustering quality on novel classes.

**Clustering Accuracy (ACC).** We adopt *ACC* as a core metric to evaluate the performance of methods. Given the predicted cluster assignments $\mathcal{G} = \{\mathcal{G}_1, \cdots, \mathcal{G}_{|\mathcal{C}|}\}$ and ground-truth $\{y_i\}_{i=1}^{N}(N = N_l + N_u)$ for the novel classes, ACC is computed by solving an optimal one-to-one mapping $\pi^*$ between cluster indices and class labels using the Hungarian algorithm (Kuhn, 1955):

$$\text{ACC} = \max_{\pi \in \mathcal{P}} \frac{1}{n} \sum_{i=1}^{N} \mathbb{I}\left[\pi(\mathcal{G}_i) = y_i\right], \tag{15}$$

where $\mathcal{P}$ denotes the set of all permutations over class indices.

However, Cluster Acc alone may not fully capture the overall GCD performance, as it ignores the balance between the accuracy of the new class and the accuracy of the old class in the algorithm, and can be trivially high when the model collapses to memorizing known data while poorly discovering new categories.

**Harmonic Mean (H-score).** To better assess the trade-off between recognition and discovery, we report the *harmonic mean* (Xian et al., 2017) of classification accuracy on Old classes and Cluster Acc on New classes (ACC_new):

$$\text{H-score} = \frac{2 \cdot \text{Old} \cdot \text{New}}{\text{Old} + \text{New}}. \tag{16}$$

This metric penalizes imbalanced performance and encourages models to perform well on both old classes and new classes. However, this formula ignores the huge difference in the number of samples belonging to the new class and the old class in the unlabeled set. Therefore, we modify it to the following:

$$\text{H-score-3} = \frac{3 \cdot \text{Old} \cdot \text{New}}{\text{Old} + 2 \cdot \text{New}}, \tag{17}$$

to enhance the ability to measure the predictive balance between the old classes and new classes.

Table 8: Performance comparison on Aircraft.

| Methods | Aircraft | | |
|---|---|---|---|
| | All | Old | New |
| GCD | 45.0 | 41.1 | 46.9 |
| DCCL | - | - | - |
| PrCAL | 52.2 | 52.2 | 52.3 |
| SimGCD | 54.2 | 59.1 | 51.8 |
| SPTNet | 59.3 | 61.8 | 58.1 |
| CMS | 56.0 | 63.4 | 52.3 |
| LegoGCD | 55.4 | 68.3 | 48.9 |
| ProtoGCD | 56.8 | 62.5 | 53.9 |
| SelEx | 57.1 | 64.7 | 53.3 |
| DebGCD | 61.7 | 63.9 | 60.6 |
| ours | 61.7 | 69.4 | 57.8 |

Table 9: Performance comparison under the inductive GCD setting across three coarse-grained benchmarks. Some results are unavailable from the original papers, and selected numbers are reproduced by CMS and PromptCAL.

| Method | CIFAR10 | | | CIFAR100 | | | ImageNet-100 | | |
|---|---|---|---|---|---|---|---|---|---|
| | All | Old | New | All | Old | New | All | Old | New |
| GCD | 82.0 | 97.3 | 66.6 | 70.1 | 76.8 | 43.5 | 79.7 | 92.7 | 66.7 |
| ORCA | - | - | - | 77.7 | 83.6 | 53.9 | 81.3 | 94.5 | 68.0 |
| SimGCD | 96.3 | 94.9 | 97.7 | 82.1 | 83.0 | 78.6 | 85.9 | 93.1 | 78.8 |
| PromptCAL | 97.3 | 96.6 | 98.1 | 81.6 | 85.3 | 66.9 | 84.8 | 94.4 | 75.2 |
| CMS | 95.3 | 97.8 | 92.9 | 80.7 | 84.4 | 65.9 | 85.7 | 95.7 | 75.8 |
| protoGCD | - | - | - | 81.6 | 82.5 | 78.0 | - | 92.8 | 78.4 |
| ours | 97.5 | 97.8 | 97.2 | 84.0 | 83.8 | 85.8 | 87.0 | 95.1 | 78.9 |

### B.3 DETAILS OF IMPLEMENTATION

Our method is implemented under a two-stage training framework: **Stage I:** We finetune multiple blocks of the backbone network. Empirically, we freeze the first $n = 4$ blocks for fine-grained datasets, and the first $n = 9$ blocks for coarse-grained datasets. The remaining blocks are finetuned using our proposed supervised objective, which emphasizes inter-class relationships. **Stage II:** We finetune only the last block across all datasets, allowing better adaptation to the discovered class structures.

For prototype learning, we adopt the *Centroids to Prototypes* strategy, where the cluster centroids are recalculated every two epochs and used to update the prototypes.

All datasets are trained for 200 epochs with a batch size of 128. The temperature coefficient $\tau$ is set to 0.1. The initial learning rate is 0.05 and is decayed to $5 \times 10^{-5}$ with a cosine schedule. All experiments are conducted on a single NVIDIA GeForce RTX 4090 GPU.

## C ADDITIONAL EXPERIMENTS

### C.1 COMPARATIVE EXPERIMENTS ON AIRCRAFT

Table 8 presents the performance of the proposed CURE on Aircraft. Our method achieves highly competitive results, tying with DebGCD for the best accuracy on All Classes, and surpassing the second-best method, SPTNet, by 2.4% on Old classes. Furthermore, CURE achieves the highest accuracy on Old Classes, reaching 69.4%. However, as shown in Table 13, CURE performs slightly worse than DebGCD in terms of class-balanced evaluation (H-score-3), indicating a slight imbalance between the old and new classes.

Table 10: Performance comparison under the inductive GCD setting on four fine-grained benchmarks. Some results are unavailable from the original papers, and selected numbers are reproduced by CMS and PromptCAL.

| Method | CUB | | | Stanford-Cars | | | Herbarium19 | | | Aircraft | | |
|---|---|---|---|---|---|---|---|---|---|---|---|---|
| | All | Old | New | All | Old | New | All | Old | New | All | Old | New |
| GCD | 57.5 | 64.5 | 57.6 | 35.4 | 50.4 | 20.9 | 39.0 | 46.7 | 30.9 | - | - | - |
| ORCA | 40.7 | 61.2 | 20.2 | - | - | - | - | - | - | - | - | - |
| SimGCD | 64.4 | 72.2 | 56.6 | 55.2 | 68.6 | 42.3 | 45.4 | 60.9 | 29.0 | - | - | - |
| PromptCAL | 62.4 | 68.1 | 56.8 | 49.1 | 63.1 | 35.7 | 40.8 | 44.7 | 36.7 | 50.1 | 56.7 | 43.4 |
| CMS | 69.7 | 76.5 | 63.0 | 57.8 | 75.2 | 41.0 | 46.2 | 53.0 | 38.9 | 53.3 | 62.7 | 43.8 |
| protoGCD | - | - | - | 55.1 | 68.9 | 41.2 | - | - | - | 57.9 | 62.1 | 53.7 |
| ours | 75.4 | 80.3 | 70.5 | 70.8 | 84.9 | 57.2 | 48.7 | 53.0 | 44.2 | 64.4 | 68.7 | 56.2 |

Table 11: Effectiveness of the logic adjustment.

| Dataset | Setting | All | Old | New |
|---|---|---|---|---|
| CUB-200 | w/o LA | 73.8 | 74.5 | 73.5 |
| | LA | 75.2 | 75.5 | 75.1 |
| CIFAR-100 | w/o LA | 85.4 | 74.7 | 86.7 |
| | LA | 85.7 | 85.3 | 86.6 |
| Stanford-Cars | w/o LA | 68.7 | 80.1 | 62.8 |
| | LA | 68.7 | 80.8 | 63.0 |

## C.2 EVALUATION ON INDUCTIVE GCD

We further compare the proposed method with several state-of-the-art approaches under the inductive setting, which is used to evaluate the generalization ability of the methods on unseen test samples, unlike the transductive setting. As shown in Table 9 and Table 10, our method consistently achieves the best performance across all benchmarks. This demonstrates that the prototypes learned by CURE through the C2P mechanism effectively capture the semantic structure of each category, thereby enabling better generalization to unseen data.

It is worth noting that in protoGCD, the original paper only reports performance on Old and New classes for four datasets, without providing results on All classes. For a more complete comparison, the All class results in the table are estimated using the following formula:

$$\text{All} = \frac{|\mathcal{C}_l|}{|\mathcal{C}|} \cdot \text{Old} + \frac{|\mathcal{C}_u|}{|\mathcal{C}|} \cdot \text{New}. \qquad (18)$$

Although this estimation may introduce minor deviations from the actual values, our empirical experience suggests that such deviations are typically small and fall within an acceptable range.

## C.3 COMPARISON WITH STATE-OF-THE-ART WITH HARMONIC MEAN

The Harmonic Mean provides a useful metric for evaluating not only the overall clustering performance across all categories, but also the prediction balance between old and novel classes. Among the datasets used in our evaluation, only CIFAR-100 maintains an approximately 1:1 ratio between the old and novel class samples; in contrast, the other datasets exhibit unbalanced distributions, with a typical ratio close to 1:2. As a result, applying the standard definition of the H-score (Eq. 16), which implicitly assumes balanced class distributions, can severely underestimate the impact of novel class performance.

For example, as shown in Table 13, although our proposed CURE method shows slightly lower balance than DebGCD on Aircraft, the standard H-score (Eq. 16) misleadingly indicates that CURE outperforms DebGCD. To better reflect the real-world class distribution, we introduce a weighted variant of the H-score (Eq. 17), which adjusts for dataset imbalance during evaluation.

Table 12: Performance comparison on H-score, H-score-3, and ACC metrics in coarse-grained benchmarks.

| Method | CIFAR-10 | | | CIFAR-100 | | | ImageNet-100 | | |
|---|---|---|---|---|---|---|---|---|---|
| | H-score-2 | H-score-3 | All | H-score-2 | H-score-3 | All | H-score-2 | H-score-3 | All |
| GCD | 92.8 | 91.2 | 91.5 | 65.7 | - | 70.8 | 76.3 | 72.6 | 74.1 |
| DCCL | 96.7 | 96.8 | 96.3 | 73.4 | - | 75.3 | 82.7 | 80.4 | 80.5 |
| PrCAL | 97.5 | 97.9 | 97.9 | 79.5 | - | 81.2 | 84.9 | 82.6 | 83.1 |
| SimGCD | 96.6 | 97.1 | 97.1 | 79.5 | - | 80.1 | 84.8 | 82.4 | 83.0 |
| SPTNet | 96.8 | 97.4 | 97.3 | 79.7 | - | 81.3 | 86.9 | 85.0 | 85.4 |
| CMS | - | - | - | 80.3 | - | 82.3 | 86.6 | 84.0 | 84.7 |
| LegoGCD | 96.4 | 97.1 | 97.1 | 81.9 | - | 81.8 | 87.9 | 85.9 | 86.3 |
| ProtoGCD | 96.7 | 97.2 | 97.3 | 81.4 | - | 81.9 | 85.6 | 83.6 | 84.0 |
| SelEx | 96.4 | 95.9 | 95.9 | 80.5 | - | 82.3 | 85.0 | 82.4 | 83.1 |
| DebGCD | 96.6 | 97.2 | 97.2 | 82.2 | - | 83.0 | 87.5 | 85.4 | 85.9 |
| ours | 97.6 | 97.5 | 97.5 | 85.9 | - | 85.7 | 88.7 | 86.7 | 87.0 |

Table 13: Performance comparison on H-score, H-score-3, and ACC metrics in fine-grained benchmarks.

| Method | CUB-200 | | | Stanford-Cars | | | Herbarium19 | | | Aircraft | | |
|---|---|---|---|---|---|---|---|---|---|---|---|---|
| | H-score-2 | H-score-3 | All | H-score-2 | H-score-3 | All | H-score-2 | H-score-3 | All | H-score-2 | H-score-3 | All |
| GCD | 52.4 | 51.1 | 51.3 | 39.4 | 35.6 | 39.0 | 35.3 | 32.0 | 35.4 | 43.8 | 44.8 | 45.0 |
| DCCL | 62.8 | 62.8 | 63.5 | 43.9 | 41.0 | 43.1 | - | - | - | - | - | - |
| PrCAL | 63.2 | 63.2 | 62.9 | 51.4 | 47.2 | 50.2 | 37.2 | 33.9 | 37.0 | 52.2 | 52.3 | 52.2 |
| SimGCD | 61.4 | 61.4 | 60.3 | 55.4 | 51.4 | 53.8 | 44.7 | 41.6 | 44.0 | 55.2 | 54.0 | 54.2 |
| SPTNet | 66.9 | 66.9 | 65.8 | 60.8 | 56.4 | 59.0 | 44.0 | 40.6 | 43.4 | 59.9 | 59.3 | 59.3 |
| CMS | 69.7 | 69.7 | 68.2 | 58.6 | 54.4 | 56.9 | 35.7 | 31.9 | 36.4 | 57.3 | 55.5 | 56.0 |
| LegoGCD | 65.3 | 65.3 | 63.8 | 59.0 | 55.0 | 57.3 | 46.0 | 43.2 | 45.1 | - | - | 18.5 |
| ProtoGCD | 64.3 | 64.3 | 63.2 | 55.3 | 51.0 | 53.8 | 45.2 | 41.9 | 44.5 | 57.9 | 56.5 | 56.8 |
| SelEx | 74.0 | 74.0 | 73.6 | 60.4 | 56.6 | 58.5 | 39.9 | 36.5 | 39.6 | 58.4 | 56.6 | 57.1 |
| DebGCD | 67.4 | 67.4 | 66.3 | 67.4 | 63.7 | 65.3 | 45.4 | 42.1 | 44.7 | 62.2 | 61.7 | 61.7 |
| ours | 75.3 | 75.3 | 75.2 | 70.8 | 68.0 | 68.7 | 48.9 | 47.7 | 48.1 | 63.0 | 61.1 | 60.2 |

Table 14: Performance on SimGCD with different numbers of tuned blocks. # Tuned Blocks indicates the index from which layers are unfrozen and updated during training.

| # Tuned Blocks | CUB-200 | | | CIFAR-100 | | |
|---|---|---|---|---|---|---|
| | All | Old | New | All | Old | New |
| 11 | 60.3 | 65.6 | 57.7 | 80.1 | 81.2 | 77.8 |
| 10 | 58.5 | 60.2 | 57.7 | 79.8 | 80.8 | 77.9 |
| 9 | 56.1 | 54.8 | 56.7 | 75.6 | 76.7 | 73.3 |
| 8 | 56.4 | 52.2 | 58.5 | 77.2 | 79.5 | 72.5 |
| 7 | 52.0 | 50.3 | 52.8 | 69.0 | 74.7 | 57.8 |
| 6 | 50.3 | 48.8 | 51.0 | 68.2 | 74.6 | 55.4 |

Table 15: Performance with different numbers of tuned blocks in the first stage of CURE. # Tuned Blocks indicates the starting block index for fine-tuning.

| # Tuned Blocks | CUB-200 | | | CIFAR-100 | | |
|---|---|---|---|---|---|---|
| | All | Old | New | All | Old | New |
| 1 | 73.7 | 73.9 | 73.5 | 85.9 | 85.1 | 87.5 |
| 2 | 74.0 | 76.1 | 72.3 | 86.0 | 85.0 | 87.9 |
| 3 | 74.5 | 75.9 | 73.9 | 85.6 | 85.4 | 85.9 |
| 4 | 74.6 | 74.5 | 74.6 | 85.7 | 85.3 | 86.6 |
| 5 | 75.2 | 76.2 | 74.7 | 85.1 | 84.5 | 86.3 |
| 6 | 73.3 | 73.1 | 73.5 | 85.2 | 84.4 | 86.6 |
| 7 | 72.9 | 72.9 | 72.9 | 85.7 | 85.0 | 87.1 |
| 8 | 72.8 | 74.3 | 72.1 | 85.5 | 84.9 | 86.6 |
| 9 | 72.1 | 71.9 | 72.1 | 85.0 | 84.9 | 85.4 |
| 10 | 71.7 | 73.3 | 70.8 | 84.3 | 84.4 | 84.1 |
| 11 | 69.8 | 70.9 | 69.3 | 83.8 | 83.7 | 83.9 |

Table 16: Effect of the semantic regularization coefficient $\lambda$. Note that for fine-grained datasets, $\lambda$ is scaled by a factor of 10 compared to coarse-grained datasets.

| $\lambda$ | CUB-200 | | | CIFAR-100 | | |
|---|---|---|---|---|---|---|
| | All | Old | New | All | Old | New |
| 0 | 73.9 | 73.9 | 73.9 | 84.8 | 85.9 | 82.8 |
| 5 / 0.5 | 74.1 | 74.1 | 74.1 | 85.5 | 85.3 | 85.9 |
| 10 / 1.0 | 74.0 | 73.8 | 74.1 | 85.9 | 85.2 | 87.3 |
| 15 / 1.5 | 74.7 | 75.7 | 74.2 | 85.6 | 85.5 | 86.0 |
| 20 / 2.0 | 75.2 | 75.5 | 75.1 | 85.2 | 85.4 | 84.9 |
| 25 / 2.5 | 73.6 | 74.7 | 73.1 | 85.8 | 85.3 | 86.8 |

Table 17: Effect of applying C2P at different iteration intervals on performance.

| epochs | CUB-200 | | | CIFAR-100 | | |
|---|---|---|---|---|---|---|
| | All | Old | New | All | Old | New |
| 1 | 75.0 | 75.7 | 74.7 | 85.9 | 85.7 | 86.3 |
| 2 | 75.2 | 75.5 | 75.1 | 85.8 | 85.6 | 86.2 |
| 3 | 74.7 | 75.7 | 74.2 | 85.9 | 85.7 | 86.3 |
| 4 | 74.5 | 75.9 | 73.8 | 85.8 | 85.8 | 85.9 |
| 5 | 72.9 | 75.4 | 71.7 | 85.6 | 85.3 | 86.2 |

From Table 12 and Table 13, it is evident that most existing state-of-the-art methods suffer substantial performance degradation on datasets with large accuracy gaps bewteen old classes and new classes, such as ImageNet-100, Stanford-Cars, and Herbarium19. In contrast, CURE achieves notably smaller drops in the H score in these datasets: only 0. 3%, 0. 7% and 0. 4%, respectively, demonstrating superior robustness and generalization. Moreover, our method achieves top performance across all seven benchmark datasets, outperforming existing state-of-art, with the sole exception being a marginal 0.6% deficit compared to DebGCD on Aircraft.

## C.4 EFFECTIVENESS OF LOGIC ADJUSTMENT.

To ensure fair comparison, we additionally evaluate a proposed logic adjustment (LA) module. As shown in Table 11, LA brings marginal improvements on CUB-200 but has negligible effect on CIFAR-100 and Stanford-Cars. We attribute this to the stronger prediction imbalance in CUB-200.

Table 18: Impact of the alignment balance factor $\alpha$ on CUB-200 and CIFAR-100. Smaller $\alpha$ emphasizes self-to-prototype alignment over full class-wise alignment.

| $\alpha$ | CUB-200 | | | CIFAR-100 | | |
|---|---|---|---|---|---|---|
| | All | Old | New | All | Old | New |
| 0.65 | 73.8 | 75.6 | 72.4 | 85.3 | 85.3 | 85.2 |
| 0.60 | 73.9 | 73.6 | 74.1 | 85.7 | 85.6 | 85.7 |
| 0.55 | 73.6 | 74.4 | 73.2 | 85.6 | 85.5 | 85.7 |
| 0.50 | 74.3 | 74.3 | 74.3 | 85.6 | 85.3 | 86.1 |
| 0.45 | 74.4 | 75.9 | 73.7 | 85.4 | 85.2 | 85.6 |
| 0.40 | 74.5 | 74.9 | 74.3 | 85.6 | 85.4 | 86.2 |
| 0.35 | 75.2 | 75.5 | 75.1 | 85.7 | 85.3 | 86.6 |
| 0.30 | 74.6 | 75.0 | 74.4 | 85.4 | 85.4 | 85.4 |

## C.5 THE IMPACT OF HYPERPARAMETERS

**Effect of Tuning Different Numbers of Transformer Blocks.** We further investigate the effect of tuning different numbers of transformer blocks on model performance. Specifically, for the first

Table 19: Ablation study of fine-tuning and contrastive learning (CL) on CUB-200 and CIFAR-100. "$\mathcal{L}_{\text{CE}}$" refers to fine-tuning with cross-entropy loss, and "$\mathcal{L}_{\text{align}}$" adds the prototype alignment loss.

| CL | Fine-tune | CUB-200 | | | CIFAR-100 | | |
|---|---|---|---|---|---|---|---|
| | | All | Old | New | All | Old | New |
| w/o CL | w/o | 61.0 | 65.0 | 59.0 | 76.5 | 77.1 | 76.2 |
| | $\mathcal{L}_{\text{CE}}$ | 66.4 | 70.9 | 64.1 | 82.5 | 86.1 | 74.6 |
| | $\mathcal{L}_{\text{CE}}+\mathcal{L}_{\text{align}}$ | 68.9 | 71.1 | 67.8 | 82.8 | 84.7 | 78.9 |
| with CL | w/o | 60.3 | 65.6 | 57.7 | 80.1 | 81.2 | 77.8 |
| | $\mathcal{L}_{\text{CE}}$ | 63.7 | 68.1 | 61.5 | 83.6 | 86.4 | 77.9 |
| | $\mathcal{L}_{\text{CE}}+\mathcal{L}_{\text{align}}$ | 64.3 | 71.0 | 61.0 | 83.7 | 86.2 | 78.8 |

stage of the proposed CURE framework, we gradually increase the number of fine-tuned blocks from one to eleven. The results are summarized in Table 15. For SimGCD, due to its dual-view consistency mechanism, the model processes $2\times$ the batch size of input images. As such, for computational efficiency, we only evaluate the effect of tuning blocks from index 11 to 6.

As shown in Table 14, SimGCD exhibits a clear performance degradation on both datasets as more blocks are fine-tuned. This drop is especially significant in terms of accuracy on old classes. For example, on CUB-200, fine-tuning only the last block yields an overall accuracy of 65.6%, while tuning up to the $6^{\text{th}}$ block results in a drop to 48.8%. This trend is likely due to the fact that the pretrained backbone provides strong general-purpose representations, where lower layers capture fundamental visual patterns. SimGCD's training strategy involves a large amount of unlabeled data, which introduces noise. As more parameters are updated, the impact of this noise becomes more pronounced, particularly in earlier layers.

In contrast, our proposed CURE consistently improves performance as more blocks are fine-tuned, as observed in Table 15. This improvement can be attributed to the fact that in the first stage of CURE, the model is trained solely on labeled data with clear class supervision. This supervised adaptation effectively aligns the pretrained backbone with the current data distribution, thereby enhancing the representation capacity of the model.

**Effect of the semantic regularization coefficient $\lambda$.** The hyperparameter $\lambda$ controls the strength of semantic regularization on prototype similarity, and the corresponding results are reported in Table 16. Empirically, coarse-grained datasets tend to exhibit weaker semantic correlations among classes, resulting in lower similarity between their associated prototypes. When $\lambda$ is set too high, the model may overemphasize semantically irrelevant but visually similar features, leading to a sharp decline in performance. Therefore, on coarse-grained datasets, we reduce $\lambda$ to one-tenth of its value used for fine-grained datasets.

Conversely, a small $\lambda$ on fine-grained datasets is insufficient to alleviate the orthogonalization of prototypes and the fragmentation of the representation space. As a result, the model fails to exploit the latent semantic relationships between classes, which is essential for improving the representation of unlabeled samples.

**Impact of C2P Update Frequency.** C2P acts as a critical bridge in CURE to connect clustering-level semantics with instance-level representations. It ensures the effective transmission of semantic information from clusters to prototypes during the second stage. As illustrated in Figure 1, the semantic regularization term encourages prototypes of different classes to aggregate, which gradually increases the distance between prototypes and instance features. By repeatedly performing C2P, the aggregated prototypes are updated back to cluster centroids, effectively re-dispersing them in the semantic space. Table 17 clearly shows that maintaining a relatively high update frequency stabilizes the algorithm's performance. In particular, executing C2P every 1, 2, or 3 epochs yields nearly identical results, demonstrating the robustness of the method to update frequency.

**Impact of the alignment balance factor $\alpha$.** To further investigate the impact of different alignment objectives in first stage of proposed CURE, we introduce a hyperparameter $\alpha$ to balance the prototype alignment loss between samples of the same class and the alignment loss between a sample and its own prototype. As shown in Table 18, we conduct a systematic ablation study on CUB-200

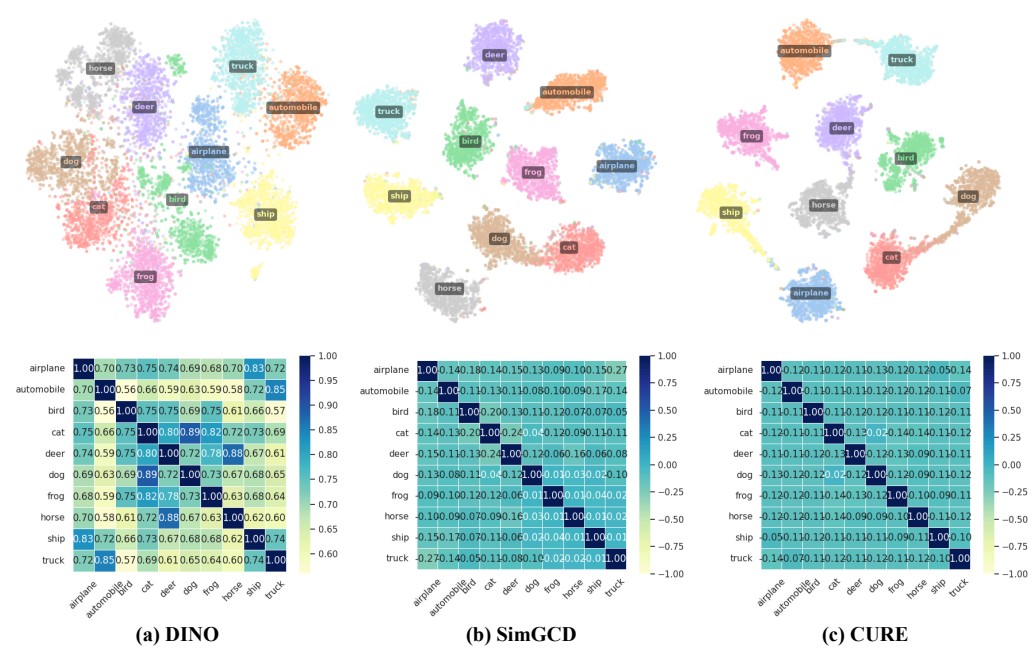

Figure 5: Visualizations of feature distribution and prototypes using DINO, SimGCD, and proposed CURE.

and CIFAR-100. The results demonstrate that reducing $\alpha$ leads to consistent improvements on All classes on both datasets, with the best performance achieved at $\alpha = 0.35$. The improvement is particularly notable on CUB-200, with the accuracy increasing from 73.8% to 75.2%. This suggests that stronger self-to-prototype alignment facilitates the learning of more discriminative features, especially for fine-grained categories. Moreover, the accuracy gain on unseen classes is more pronounced, confirming that enhancing self-to-prototype alignment significantly benefits generalization to novel categories. Therefore, we adopt $\alpha = 0.35$ as the default setting in our final model.

Table 20: Estimated category numbers and ground-truth category numbers in the unlabeled data.

| Method | CUB-200 | Stanford-Cars | CIFAR-100 | ImageNet-100 |
|---|---|---|---|---|
| Ground-truth $K$ | 200 | 196 | 100 | 100 |
| Estimated $K$ | 231 | 230 | 100 | 109 |

Table 21: Performance comparison on four benchmark datasets with the estimated number of categories.

| Method | CUB | | | Stanford-Cars | | | CIFAR-100 | | | ImageNet-100 | | |
|---|---|---|---|---|---|---|---|---|---|---|---|---|
| | All | Old | New | All | Old | New | All | Old | New | All | Old | New |
| GCD | 47.1 | 55.1 | 44.8 | 35.0 | 56.0 | 24.8 | 73.0 | 76.2 | 66.5 | 72.7 | 91.8 | 63.8 |
| SimGCD | 61.5 | 66.4 | 59.1 | 49.1 | 65.1 | 41.3 | 80.1 | 81.2 | 77.8 | 81.7 | 91.2 | 76.8 |
| $\mu$GCD | 62.0 | 60.3 | 62.8 | 56.3 | 66.8 | 51.1 | - | - | - | - | - | - |
| DebGCD | 64.5 | 68.5 | 62.5 | 63.3 | 78.6 | 55.8 | 83.0 | 84.6 | 79.9 | 84.9 | 93.3 | 80.7 |
| SelEx | 72.0 | 72.3 | 71.9 | 58.7 | 75.3 | 50.8 | - | - | - | - | - | - |
| Ours | 73.7 | 74.1 | 73.6 | 67.2 | 80.5 | 60.8 | 85.7 | 85.3 | 86.6 | 84.7 | 95.1 | 79.6 |

Table 22: Split of labeled and unlabeled classes in CIFAR-10.

| Dataset | $\mathcal{C}_l$ | $\mathcal{C}_u$ |
|---|---|---|
| CIFAR-10 | 'airplane', 'automobile', 'bird', 'cat', 'deer' | 'dog', 'frog', 'horse', 'ship', 'truck' |

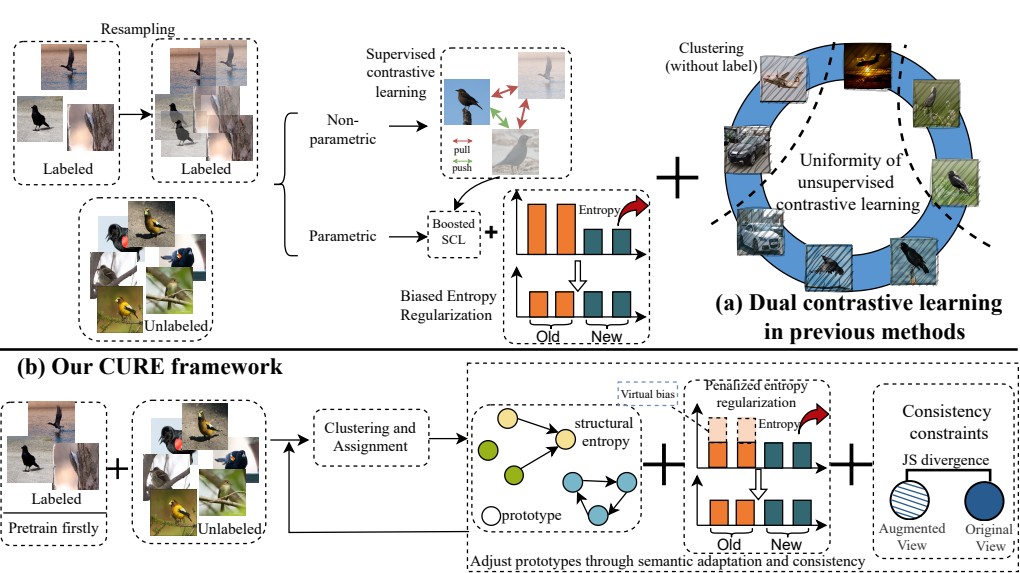

Figure 6: Conceptual comparison between previous dual CL paradigms and our proposed CURE framework. Subfigure (a) illustrates the limitations of existing methods, which rely heavily on augmented views and biased entropy. Subfigure (b) shows our unified framework CURE, which integrates semantic prototype adaptation, entropy regularization, and view-level consistency for discovering novel classes.

## C.6 THE ABLATION STUDY OF FINE-TUNING STRATEGIES AND CL

To further assess the effectiveness of fine-tuning strategies and the contribution of CL to representation quality, we conduct a detailed ablation study under six different configurations. As shown in Table 19, we compare results with and without CL, and examine the impact of applying no fine-tuning, fine-tuning with cross-entropy loss, and fine-tuning with both CE and proposed alignment losses.

Interestingly, the results in Table 19 reveal a nuanced trend regarding the effect of different fine-tuning strategies. On the one hand, fine-tuning the model using labeled data significantly improves performance across datasets. On the other hand, CL exhibits a bifurcated impact: while it severely degrades performance on fine-grained datasets, it yields moderate improvements on coarse-grained datasets.

This observation may be attributed to the inherent differences in inter-class boundaries. In coarse-grained datasets, the semantic distinctions between categories are relatively clear-cut, allowing CL to enhance representation quality without substantially increasing inter-class confusion. In contrast, fine-grained datasets contain highly similar classes, where CL may disrupt intra-class consistency, leading to over-separation of semantically close samples and a sharp decline in performance.

## C.7 CATEGORY DISCOVERY WITH THE ESTIMATED NUMBER OF CATEGORIES

**Performance with Estimated Number of Categories.** Following the common practice in existing literature, we conduct our main experiments using the ground-truth number of categories. In this section, we report the performance of CURE when using the estimated number of categories, derived from the off-the-shelf method proposed by Vaze *et al.* (Vaze et al., 2022a), to simulate sce-

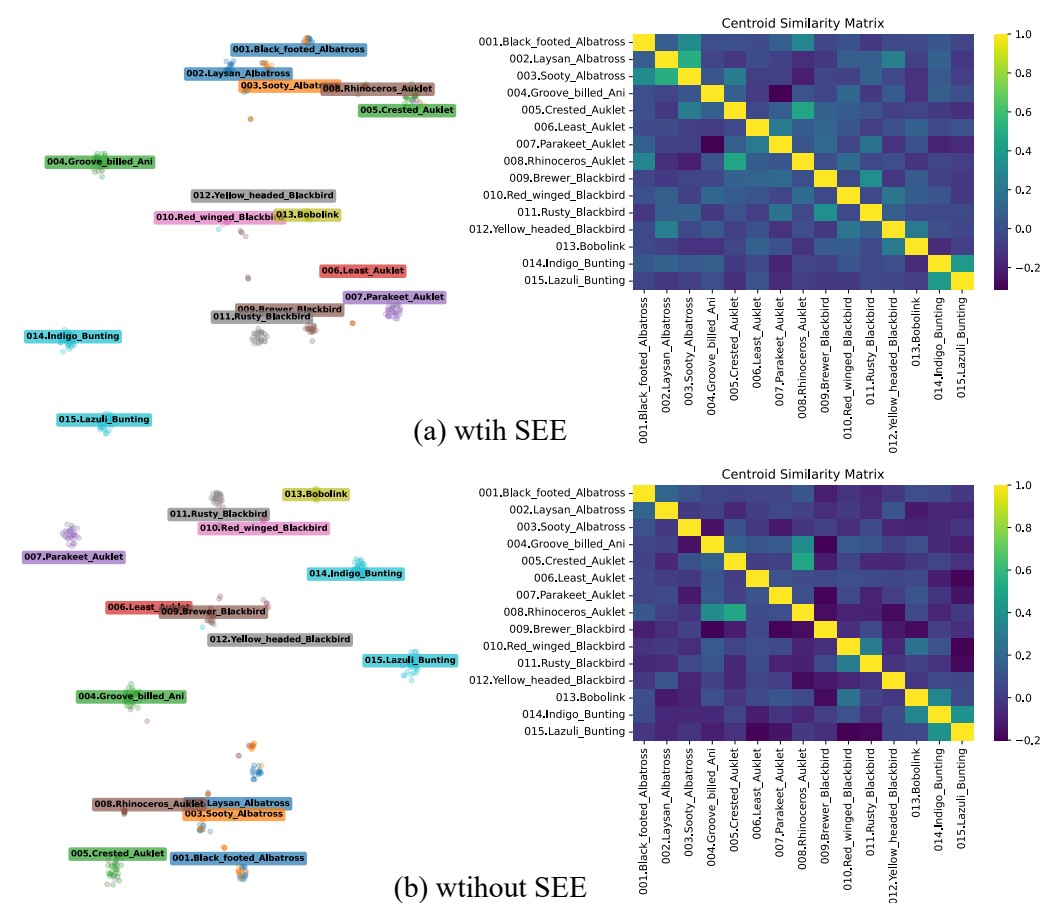

(a) wtih SEE

(b) wtihout SEE

Figure 7: t-SNE and Prototype Similarity Under SEE vs. w/ SEE. t-SNE visualizations show that SEE enhances inter-class separation while preserving fine-grained semantic neighborhoods. Prototype similarity heatmaps indicate that SEE avoids prototype space collapse and maintains meaningful semantic relationships.

narios where the true number of categories is unavailable. Table 20 reports the estimated category numbers used for evaluation. In Table 21, we compare CURE with SimGCD (Wen et al., 2023), $\mu$GCD (Vaze et al., 2023), GCD (Vaze et al., 2022a), DebGCD (Liu & Han, 2025), and SelEx (Rastegar et al., 2024) under this setting. Notably, despite a gap of approximately 15% between the estimated and ground-truth number of classes on CUB-200 and Stanford-Cars, our method exhibits significantly smaller performance degradation. On CUB-200, for instance, the degradation is less than 2%. Even when using the same estimated category number across all four datasets, CURE remains the most competitive method on All Classes, outperforming others on three datasets and falling behind DebGCD by only 0.2% on ImageNet-100.

# D VISUALIZATION

## D.1 VISUALIZING FEATURE REPRESENTATIONS ACROSS METHODS

To qualitatively evaluate the learned representations of unlabeled data, we visualize the feature distributions generated by different methods using t-SNE in a 2D space, as shown in Figure 5. The representations produced by DINO exhibit highly Unclear distribution with indistinct class boundaries. SimGCD partially improves the separability, but still suffers from significant cluster overlap. In contrast, our proposed method, CURE, demonstrates substantially better intra-class compactness

and inter-class separability, indicating more discriminative and semantically consistent representations for novel classes.

Furthermore, it is evident from Figure 5 that CURE preserves semantic relations among similar categories. Based on the novel/known class partition in Table 22, we observe a strong semantic connection between the novel class *dog* and the known class *cat*, suggesting that our semantic regularization plays an important role in shaping class-level relations and enabling novel categories to benefit from the semantic structure encoded by known prototypes.

## D.2 Conceptual Comparison with Existing State-of-art

Figure 6 presents a conceptual comparison between traditional dual CL methods and our proposed CURE framework. As shown in subfigure (a), existing approaches typically involve CL between labeled and unlabeled data, combined with resampling, parametric/non-parametric objectives, or entropy regularization. These methods all rely on self-supervised CL and its variants to represent unlabeled data, resulting in features that are not friendly to clustering.

In contrast, as illustrated in subfigure (b), our CURE framework introduces a unified strategy that combines prototype-based semantic adaptation, entropy-aware bias penalization, and view-level consistency constraints. This design enables a more coherent and structure-preserving learning process, leading to improved generalization on unseen categories. Our method completely avoids this self-supervised CL by learning the category relationships of known categories, then relying on the learned semantic structure to expand new categories, refining the prototype representation of new categories through semantic regularization terms, and finally converting them into representations of unlabeled data.

## D.3 Visualization of Semantic Structure Preservation

To further examine whether SEE preserves semantic structure as intended, we conduct qualitative visualizations on the fine-grained CUB-200 dataset. We select 15 representative bird categories and present two complementary forms of analysis: t-SNE embeddings and prototype–prototype similarity heatmaps.

We extract image features and project them into a 2D space using t-SNE. As shown in Figure 7, the model trained without SEE exhibits notable feature collapse: semantically related species become dispersed or entangled, losing neighborhood structure. In contrast, the version with SEE presents clearer inter-class separation while still maintaining meaningful proximity between visually similar bird species. This directly indicates that SEE helps preserve fine-grained semantic connection.

To obtain reliable class prototypes, we first perform clustering on the extracted features to suppress noise. We then match cluster assignments with ground-truth labels and retain only correctly predicted samples for computing class centers. The resulting similarity matrices are shown in Figure 7. Without SEE, the prototype similarity map degenerates toward a near-orthogonal structure, suggesting a collapse of semantic relations. With SEE, however, semantically related categories maintain strong off-diagonal connections, forming meaningful local blocks that reflect true visual similarity. This confirms that SEE prevents prototype space shrinkage and preserves relational structure across fine-grained classes.

Together, these visualizations provide strong evidence that SEE effectively maintains semantic consistency while avoiding the over-separation or orthogonalization observed in baseline models.

