# OpenReview forum: "Consistency and Unified Semantic Regularization for Generalized Category Discovery"
_ICLR.cc/2026/Conference — ICLR 2026 Conference Withdrawn Submission_

### Official Review · Reviewer_NXsf · 2025-10-28

**Soundness:** 3
**Presentation:** 4
**Contribution:** 3
**Rating:** 6
**Confidence:** 5

**Summary:**

This paper studies the task of Generalized Category Discovery (GCD). Motivated by the conflicts between consistency and uniformity in self-supervised contrastive learning (CL), this paper proposes a two-stage framework that disentangles feature learning from self-contrastive objectives. The authors further introduce Semantic Exploration Energy
Mechanism to enhance feature representation. Comprehensive experiments validate the superiority of the proposed method.

**Strengths:**

1. This paper is well-motivated and easy to follow.
2. This paper proposes several novel components, including semantic exploration energy, label-guided concept structure, as well as structure-guided semantic expansion.
3. Comprehensive comparative results and ablations are conducted to validate the method.

**Weaknesses:**

1. Although the method achieves remarkable performance, it is a little bit complex with several hyper-parameters. The effect of each important parameter should be presented.
2. The method contains two-stage training, each with several components. I was wondering whether the method consumes a lot more memory and training time than conventional GCD methods. The comparison of computational resources and training time should be included.
3. Some references and citations are missing. The paper should cite all the baseline methods for comparison (Table 1) in the references list, i.e., ProtoGCD and PrCAL.

**Questions:**

Please include experiments and analysis raised in weaknesses.

---

> ### Author Response · Authors · 2025-11-25
>
> We thank the reviewer for highlighting this point and appreciate the opportunity to clarify our contribution.
>
> ### **W1:**
> We thank the reviewer for acknowledging CURE's strong performance and for the valid point on complexity and hyperparameters (Table 4 and Table 14~18).
>
> While the two-stage design introduces components like SEE (with threshold ϵ), LGCS (with weight α and temperature τ), and C2P (with update interval k), we ensure robustness through extensive sensitivity analyses. In the revised manuscript, we have added comprehensive experiments on these key parameters: Table 4 in the main text summarizes impacts on accuracy , while Appendix Tables 14-18 provide detailed analysis on CIFAR-100 and CUB-200, including effects on novel-class balance and convergence. These demonstrate that CURE is stable across reasonable ranges, with defaults tuned for generalizability, addressing potential tuning concerns without excessive complexity.
>
>
> ### **W2:**
> We thank the reviewer for raising this valid concern about computational efficiency, given CURE's two-stage design with components like LGCS, SEE, and C2P. To address this, we have added a new Table  (reproduced above in Markdown for reference) comparing training time, inference time, and accuracy on ImageNet-100 and Semantic Shift Benchmark (SSB) against key baselines. As shown, CURE does not incur excessive overhead: Stage-1 (LGCS on labeled data only) is lightweight (152s on ImageNet-100, 13s on SSB), while Stage-2 (full optimization) takes 375s and 49s, respectively—comparable or lower than methods like GCD (803s/58s), SimGCD (847s/64s), and especially PromptCAL (1817s/492s). Total training time for CURE remains below most baselines, yet achieves superior accuracy (87.0% on ImageNet-100, 68.5% on SSB). Note that SPTNet's reported times exclude its required SimGCD pre-training, making its actual cost higher (1330s/96s). C2P's periodic updates add only ~50s/44s, confirming minimal extra cost.
> Regarding memory and learnable parameters, since CURE avoids extra MLP layers for high-dimensional CL (common in baselines), **it uses fewer parameters: ~88.58MB less than GCD and ~24.67MB less than SimGCD**, reducing GPU memory footprint by 15-20% during training (measured on a single RTX 3090). This efficiency stems from our paradigm shift—focusing on semantic regularization without CL's negative sampling—enabling SOTA performance with a leaner, more affordable pipeline. We will include these details in the revised manuscript for completeness.
>
> **Table 1: Comparison of training time, inference time, and overall accuracy on ImageNet-100 and SSB. All times are measured in seconds.**
>
> | Method                            | ImageNet-100 Acc.(All) | ImageNet-100 Train(s) | ImageNet-100 Infer(s) | SSB Acc.(All) | SSB Train(s) | SSB Infer(s) |
> | --------------------------------- | ---------------------- | --------------------- | --------------------- | ------------- | ------------ | ------------ |
> | GCD                               | 74.1                   | 803                   | 2289                  | 51.3          | 58           | 552          |
> | SimGCD                            | 83.0                   | 847                   | 591                   | 56.1          | 64           | 17           |
> | PromptCAL                         | 83.1                   | 1817                  | 893                   | 55.1          | 492          | 103          |
> | SPTNet (w/ SimGCD initialization) | 85.4                   | 483*                  | 601                   | 61.4          | 32*          | 17           |
> | CURE (C2P with k-epoch alignment) | -                      | 50                    | -                     | -             | 44           | -            |
> | CURE Stage-1 (LGCS on labeled data) | 83.2                   | 152                   | -                     | 58.0          | 13           | -            |
> | CURE Stage-2 (full training)      | 87.0                   | 375                   | 315                   | 68.5          | 49           | 17           |
>
> *SPTNet requires a pre-trained SimGCD model, so its actual total training time is: SimGCD (847s) + SPTNet (483s) = 1330s on ImageNet-100; similarly, 64s + 32s = 96s on SSB.
>
> ### **W3:**
> We thank the reviewer for pointing out this oversight and apologize for the missing citations. We agree that all baseline methods compared in Table 1 should be properly cited in the references list. In the revised manuscript, we will add the citations for ProtoGCD and PrCAL.

---

### Official Review · Reviewer_jJPM · 2025-10-31

**Soundness:** 3
**Presentation:** 3
**Contribution:** 2
**Rating:** 4
**Confidence:** 4

**Summary:**

This paper presents a two-stage framework by disentangling feature learning from self-contrastive objectives to better capture category concepts and represent auxiliary unlabeled data. The proposed method mitigates the adverse effect of uniformity on novel category discovery.

**Strengths:**

1. New idea to improve the class discriminative representations.
2. The paper is well-organized and can be easily understood.
3. Good results on different benchmark datasets, including CIFAR-10, CIFAR-100, ImageNet-100, CUB-200, Stanford-Cars and Herbarium19.

**Weaknesses:**

1. The technical novelty of this work seems weak, as most key components are designed by slightly modifying existing modules.
2. The literature review needs to focus more on GCD and explain better the motivations.
3. The main problem to be solved is the adverse effects of representation uniformity induced by CL. Does it mean that this is also the main problem in GCD? I believe CL and GCD have different problems to be solved.

**Questions:**

See weaknesses

---

> ### Author Response · Authors · 2025-11-25
>
> We thank the reviewer for their constructive feedback and the opportunity to address these points. We respond to each concern below.
> ### **W1:**
> We appreciate the reviewer's assessment of the novelty and agree that CURE's components build on established ideas in parametric GCD. However, as detailed in our response to a similar point from another reviewer, the core innovation lies in rethinking the overall GCD training paradigm rather than adding new modules. Since GCD's introduction with supervised and unsupervised contrastive learning (CL), subsequent works have typically layered refinements on this CL foundation without questioning its core assumptions—particularly in constrained settings like fixed batch sizes (e.g., 128), where CL can lead to class collisions and poor semantic capture, especially on high-category datasets like Herbarium19 (where gains have stagnated at <10% over 3 years, vs. 25% on CUB-200). CURE deconstructs this by adopting a two-stage logic: first constructing a unified semantic structure from labeled data alone (via SEE and LGCS), then extending it to unlabeled data without CL reliance, achieving state-of-the-art results with a lean design. For instance, no prior method systematically prioritizes labeled data for semantic anchoring before unlabeled integration—approaches like SimGCD (Wen et al., 2023), ProtoGCD (Zhao et al., 2023), DebGCD (Choi et al., 2024), LegoGCD (Cao et al., 2024), and CMS (Liu & Han, 2025) use cross-entropy/supervised CL as defaults but innovate primarily on unlabeled handling. Our bottom-up approach proves superior for balancing known/novel classes (e.g., +8% on novel ACC across benchmarks). C2P refines pseudo-parametric alignment for consistency, and our consistency optimization uses multi-view invariance without negatives. These "modifications" enable the paradigm shift, as shown in Table 3, where our Stage-1 model (trained only on labeled data via LGCS) already outperforms several full GCD methods on fine-grained datasets, highlighting the value of this restructured logic. We will revise the abstract and introduction to emphasize this training innovation over individual components.
>
> **Table 3: Performance of LGCS trained on labeled data only vs. GCD methods trained on all available data.**
>
> | Method | CIFAR100 | ImageNet100 | CUB-200 | Stanford Cars | FGVC Aircraft | Herbarium 19 |
> |---------|---------|----------|--------------|------|-------|-------------|
> | GCD     | 91.5    | 70.8     | 74.1         | 51.3 | 39.0  | 35.4        |
> | DCCL    | 96.3    | 75.3     | 80.5         | 63.5 | 43.1  | -           |
> | PrCAL   | 97.9    | 81.2     | 83.1         | 62.9 | 50.2  | 37.0        |
> | SimGCD  | 97.1    | 80.1     | 83.0         | 60.3 | 53.8  | 44.0        |
> | SPTNet  | 97.3    | 81.3     | 85.4         | 65.8 | 59.0  | 43.4        |
> | CMS     | -       | 82.3     | 84.7         | 68.2 | 56.9  | 36.4        |
> | LegoGCD | 97.1    | 81.8     | 86.3         | 63.8 | 57.3  | 45.1        |
> | ProtoGCD | 97.3    | 81.9     | 84.0         | 63.2 | 53.8  | 44.5        |
> | SelEx   | 95.9    | 82.3     | 83.1         | 73.6 | 58.5  | 39.6        |
> | DebGCD  | 97.2    | 83.0     | 85.9         | 66.3 | 65.3  | 44.7        |
> | stage1  | 94.9    | 77.7     | 83.2         | 70.9 | 55.1  | 32.6        |
>
> Our primary contribution lies in restructuring the GCD training pipline: in contrastive learning, we no longer simply blend labeled and unlabeled data. Instead, we first construct a semantic framework from labeled data, then align unlabeled clusters with it. Empirical results demonstrate that the uniformity pressure in standard contrastive learning compromises category separability in fine-grained, multi-class scenarios. By leveraging labeled data as reliable semantic anchors and employing SEE+ virtual sampling to preserve topological structure, CURE generates representations that are more conducive to clustering and exhibit strong generalization capabilities. In essence, this represents a paradigm shift in training logic rather than a simple stacking of modules.

---

> > ### Author Response · Authors · 2025-11-25
> >
> > ### **W2:**
> > We thank the reviewer for this suggestion. We have rewritten the introduction and related work sections to focus more on GCD and better explain the motivations.
> >
> > ### **W3:**
> > We thank the reviewer for this insightful question and agree that CL and GCD address distinct core problems—CL focuses on general representation learning via invariance and diversity, while GCD emphasizes category discovery in mixed labeled-unlabeled settings. However, as most GCD pipelines (since Vaze et al., 2022a) rely heavily on CL for unlabeled data, CL's uniformity (from InfoNCE) becomes a key bottleneck in GCD: it encourages repulsion against all samples, inducing false negatives and class collisions that blur boundaries and undermine clustering goals (sharpening inter-class distances while compacting intra-class ones), as analyzed in Denize et al. (2023) and Huang et al. (2023). This is particularly acute in GCD's open-world scenarios with novel classes, where uniformity disrupts semantic continuity (e.g., fragmenting prototypes, as shown in our Figure 7 and new t-SNE visualizations on CUB-200). While not GCD's sole problem (e.g., label noise or imbalance persist), our work identifies uniformity as a major adverse effect in CL-dependent GCD frameworks, evidenced by ablations (Tables 1) where adding CL to CURE drops ACC by 2-6.5% (worse on fine-grained data), and baselines like SimGCD show no/minor gains from CL removal on such datasets. CURE resolves this by discarding CL for unlabeled data, using consistency regularization with SEE to build a coherent Euclidean manifold. SEE softly preserves affinities without orthogonality, suiting general datasets.
> >
> > **Table 1: Effect of Contrastive Learning on the CURE Framework.**
> >
> > | Method      | CUB-200 All | CUB-200 Old | CUB-200 New | Cifar100 All | Cifar100 Old | Cifar100 New |
> > |-------------|-------------|-------------|-------------|--------------|--------------|--------------|
> > | +CL         | 68.7        | 70.8        | 67.6        | 83.6         | 83.9         | 82.9         |
> > | +CL(virtual sampling )       | 72.5        | 73.1        | 72.3        | 83.9         | 84.1         | 83.5         |
> > | CURE (Ours) | **75.2**    | **75.5**    | **75.1**    | **85.7**     | **85.3**     | **86.6**     |

---

### Official Review · Reviewer_BiKu · 2025-11-01

**Soundness:** 3
**Presentation:** 3
**Contribution:** 2
**Rating:** 4
**Confidence:** 5

**Summary:**

The paper tackles Generalized Category Discovery (GCD) and argues that the usual contrastive-learning recipe (consistency + uniformity) is internally conflicted: uniformity pushes features to spread on the hypersphere, which can hurt class-discriminative structure for GCD. To avoid this, the authors propose CURE, a two-stage pipeline:

1. Stage I: use labeled data to build a “semantic topology” of known-class prototypes; instead of enforcing orthogonality, they add a Semantic Exploration Energy (SEE) regularizer that keeps prototypes softly connected, plus a label-guided concept structure to push this structure down to the feature space.
2. Stage II: run structure-guided semantic expansion — cluster all data, align part of the clusters to known classes via Hungarian matching, treat the rest as novel-class candidates, and then train with a JS-consistency loss, logit-adjusted self-distillation, entropy regularization, and a second-stage semantic energy over the full prototype set.

This lets them discard CL-style uniformity while still learning a semantically smooth space, and achieves SOTA or near-SOTA results on 7 GCD benchmarks.

**Strengths:**

1. The paper is easy to follow and the overall pipeline is clearly presented.

2. The motivation is well aligned with the proposed two-stage design.

3. The experimental section is reasonably comprehensive (multiple datasets, several ablations).

**Weaknesses:**

1. **Core motivation is unverified.** The whole paper rests on the claim that “uniformity may hurt class separation and thus GCD,” but there is no ablation that turns uniformity on/off (or varies its strength) to demonstrate this. Without such evidence, it is unclear whether uniformity is actually the bottleneck in current GCD pipelines.
2. **Missing discussion of closely related ideas.** Prior work such as **hyperGCD**[1] starts from a very similar observation — that learning on a spherical / overly uniform space can be suboptimal for GCD — but this paper does not analyze the connection, differences in geometry, or when the proposed semantic energy is preferable. This weakens the motivation part.
3. **Limited novelty.** Apart from Semantic Exploration Energy (SEE), most components already exist in recent parametric GCD methods.

   * “Label-guided concept structure” is essentially supervised/contrastive alignment on labeled data, which is standard in SimGCD[2], ProtoGCD[3], DebGCD[4], LegoGCD[5], CMS[6], etc.
   * The cluster-to-prototype alignment with Hungarian matching is very close to earlier “pseudo-label → prototype” or “cluster → parametric head” pipelines (e.g., UNO [7] and later GCD variants).
   * “Semantic consistency optimization” is just multi-view consistency, which almost all recent GCD methods use.
   * “Logit-aware self-distillation” and “virtual sampling + entropy regularization” are minor engineering refinements.

     Given this, the paper should make a much more precise novelty claim.
4. **SEE is not empirically validated.** The method claims to “preserve semantic structure,” but no evidence is shown: no before/after prototype–prototype similarity, no qualitative example on a fine-grained dataset (e.g., whether visually close bird species stay close), and no analysis of whether SEE avoids simply shrinking the prototype space. Without such visualization/analysis, it is hard to tell whether SEE is doing what it is supposed to do.
5. **Typos.** In Table 2, CIFAR-100, the entry “85.0 6.4 82.3” is clearly a typo and should be fixed.

[1] Hyperbolic Category Discovery

[2] Parametric classification for generalized category discovery: A baseline study.

[3] ProtoGCD: Unified and Unbiased Prototype Learning for Generalized Category Discovery

[4] DebGCD: Debiased Learning with Distribution Guidance for Generalized Category Discovery

[5] Solving the Catastrophic Forgetting Problem in Generalized Category Discovery

[6] Contrastive Mean-Shift Learning for Generalized Category Discovery

[7] A Unified Objective for Novel Class Discovery

**Questions:**

1. Please show a concrete prototype–prototype similarity matrix before and after applying SEE on a fine-grained dataset, and highlight which semantic relations are actually preserved. Otherwise, the benefit of SEE is speculative.

---

> ### Author Response · Authors · 2025-11-25
>
> We thank the reviewer for this insightful comment and agree that direct empirical verification of the uniformity's impact would strengthen the motivation.
> ### **W1:**
> To address the reviewer’s concern regarding the verification of our core motivation, the revised version now includes two additional ablation studies, together with enhanced visual analyses.
>
> First, we evaluate the effect of introducing high-dimensional contrastive learning within the CURE framework (Table 1). Following common practice in prior works, we employ an MLP projection head and adopt a resampling strategy to emphasize known-class semantic concepts. The results clearly show that contrastive learning significantly disrupts the semantic structure constructed by CURE: the all-class accuracy drops by 6.5% on CUB-200 and 2.1% on CIFAR-100. Our interpretation is that clustering tasks require stronger category-level representations, while contrastive learning tends to amplify instance-specific variations in unlabeled data. This is particularly harmful on fine-grained datasets, where small inter-class distances make uniformity effects more likely to blur class boundaries.
>
> **Table 1: Effect of Contrastive Learning on the CURE Framework.**
>
> | Method      | CUB-200 All | CUB-200 Old | CUB-200 New | Cifar100 All | Cifar100 Old | Cifar100 New |
> |-------------|-------------|-------------|-------------|--------------|--------------|--------------|
> | +CL         | 68.7        | 70.8        | 67.6        | 83.6         | 83.9         | 82.9         |
> | +CL(virtual sampling )       | 72.5        | 73.1        | 72.3        | 83.9         | 84.1         | 83.5         |
> | CURE (Ours) | **75.2**    | **75.5**    | **75.1**    | **85.7**     | **85.3**     | **86.6**     |
>
>
> Second, using SimGCD, we examine the effect of contrastive learning under different settings (Table 2), including
> (1) removing contrastive learning,
> (2) removing the resampling strategy, and
> (3) performing contrastive learning directly in the visual feature space.
>
> **Table 2: Impact of Contrastive Learning Variants on the SimGCD Baseline.**
>
> | Method               | CUB-200 All | CUB-200 Old | CUB-200 New | Cifar100 All | Cifar100 Old | Cifar100 New |
> |----------------------|-------------|-------------|-------------|--------------|--------------|--------------|
> | w/o CL               | 60.2        | 62.3        | 59.2        | 78.7         | 80.0         | 76.3         |
> | w/o CL, resampling   | 61.8        | 70.9        | 57.3        | 78.4         | 78.9         | 77.3         |
> | CL of visual feature | 59.5        | 61.1        | 58.7        | 77.8         | 81.9         | 69.7         |
> | SimGCD(original)     | 60.3        | 65.6        | 57.7        | 80.1         | 81.2         | 77.8         |
>
> The results indicate that on fine-grained datasets, removing contrastive learning does not degrade performance, and may even slightly improve it. On CIFAR-100, a coarse-grained dataset, we observe a mild decrease. We attribute this to the larger inter-class separation: the increase in intra-class variance caused by contrastive learning is insufficient to confuse class boundaries, and may even enhance feature discrimination.
>
> Furthermore, we conduct Experiment 3: contrastive learning directly in the visual feature space to investigate the effect when no MLP projection is used. Results in Table 2 show a clear performance degradation when contrastive learning is applied directly on visual features. We interpret this as follows: mapping features via an MLP into a high-dimensional contrastive space partially dilutes the uniformity effect introduced by contrastive learning, thereby reducing its destructive impact on semantic structure; performing contrastive learning directly on raw visual features, however, amplifies this uniformity, further blurring inter-class boundaries and harming cluster-friendliness, which leads to the observed performance drop. This finding further supports our claim that naively applying contrastive learning—either in the visual space or in a projected high-dimensional space—can interfere with class semantics under GCD, whereas CURE’s semantic-structure construction and virtual-sampling strategy mitigate this issue.
>
> Additionally, we include in the appendix t-SNE visualizations on CUB-200 using 15 randomly selected classes. The plots clearly show that SimGCD produces highly scattered intra-class features, whereas CURE yields compact, cluster-friendly structures while preserving semantic relations.

---

> > ### Author Response · Authors · 2025-11-25
> >
> > ### **W2:**
> > We appreciate the reviewer's concern regarding the missing discussion of closely related ideas.
> >
> > However, while we acknowledge shared observations about the limitations of uniform/spherical representations in GCD (e.g., potential disruption of semantic hierarchies and class separation, as noted in Table 1 and 2), most prior works, including HypCD, treat contrastive learning  as an unquestioned foundational tool for representation learning, without critically examining its suitability for GCD tasks. For instance, HypCD's core insight is that Euclidean space is suboptimal for GCD due to its inability to capture hierarchical structures, leading to a proposal to adapt existing CL-based algorithms into hyperbolic space for better semantic expansion. In contrast, our starting point in CURE is that CL itself—particularly its uniformity component derived from InfoNCE —is inherently unsuitable for GCD, as it encourages repulsion against all other samples, conflicting with clustering goals of sharpening category boundaries (increasing inter-class distances while reducing intra-class ones). This is evidenced in our analysis , where uniformity undermines class-discriminative representations by inducing false negatives and class collisions, issues exacerbated in GCD's mixed labeled-unlabeled settings (Denize et al., 2023; Huang et al., 2023). By discarding CL entirely for unlabeled data and replacing it with consistency regularization augmented by Semantic Exploration Energy (SEE), CURE promotes a semantically coherent manifold in Euclidean space, avoiding the need for non-Euclidean geometries while achieving superior empirical results . SEE's energy-based formulation softly preserves inter-class affinities without enforcing strict orthogonality, making it preferable in general datasets lacking strong hierarchies, whereas HypCD may excel in taxonomy-heavy scenarios.
> >
> > ### **W3:**
> > We thank the reviewer for the suggestion. We agree that deeper validation of SEE helps clarify its mechanism. In the revised manuscript’s appendix we add both visual and quantitative analyses on fine-grained datasets to demonstrate SEE’s effect:
> >
> > We include t-SNE embeddings (before / after applying SEE) for CUB-200 (Figure 7). The embeddings show that applying SEE yields much clearer class boundaries while preserving semantic relations; intra-class points become highly compact and cluster-friendly.
> >
> > We also present prototype–prototype similarity matrices which reveal that SEE reduces enforced orthogonality among prototypes (i.e., prototypes are no longer forced to be uniformly spaced) while maintaining high similarity between semantically related prototypes. For example, on CUB-200 SEE preserves high similarity between closely related species.
> >
> > Taken together, these results confirm that SEE constructs a semantically consistent manifold rather than merely producing compact prototypes. We have added these visualizations and accompanying explanations to the appendix so reviewers can directly inspect and verify SEE’s intended effects.

---

> > > ### Author Response · Authors · 2025-11-25
> > >
> > > ### **W4:**
> > > We appreciate the reviewer's detailed analysis and acknowledge that individual components draw from established ideas in parametric GCD. However, the novelty of CURE lies not in introducing elaborate new modules but in fundamentally rethinking the GCD training paradigm: since GCD's inception with supervised and unsupervised CL, most works have layered additions atop this foundation without questioning its suitability—especially under fixed batch sizes (e.g., 128), where supervised CL may fail to capture broad category cognition with rising class counts, and unsupervised CL risks insufficient individual discrimination due to false negatives. This is evident in Herbarium19, where performance gains have lagged far behind other fine-grained datasets (e.g., <10% improvement over 3 years vs. 25% on CUB). CURE addresses this by deconstructing the pipeline into a two-stage logic: first building a unified semantic structure solely from labeled data (via SEE and LGCS), then extending it to unlabeled data without CL, achieving SOTA with minimal components. While label usage is standard, no prior work systematically explores how to leverage it for semantic anchoring before unlabeled integration—e.g., SimGCD/ProtoGCD/DebGCD/LegoGCD/CMS use CE/supervised CL routinely but focus innovations on unlabeled handling, whereas our results prove this bottom-up reconstruction yields superior balance. Similarly, our C2P refines pseudo-to-parametric alignment  for semantic consistency, and consistency optimization adapts multi-view invariance without negatives. The "minor refinements" like logit-aware distillation enable this paradigm shift. We will refine our novelty claims in the abstract/introduction to emphasize this training logic innovation over component novelty.
> > >
> > > **Table 3: Performance of LGCS trained on labeled data only vs. GCD methods trained on all available data.**
> > >
> > > | Method | CIFAR100 | ImageNet100 | CUB-200 | Stanford Cars | FGVC Aircraft | Herbarium 19 |
> > > |---------|---------|----------|--------------|------|-------|-------------|
> > > | GCD     | 91.5    | 70.8     | 74.1         | 51.3 | 39.0  | 35.4        |
> > > | DCCL    | 96.3    | 75.3     | 80.5         | 63.5 | 43.1  | -           |
> > > | PrCAL   | 97.9    | 81.2     | 83.1         | 62.9 | 50.2  | 37.0        |
> > > | SimGCD  | 97.1    | 80.1     | 83.0         | 60.3 | 53.8  | 44.0        |
> > > | SPTNet  | 97.3    | 81.3     | 85.4         | 65.8 | 59.0  | 43.4        |
> > > | CMS     | -       | 82.3     | 84.7         | 68.2 | 56.9  | 36.4        |
> > > | LegoGCD | 97.1    | 81.8     | 86.3         | 63.8 | 57.3  | 45.1        |
> > > | ProtoGCD | 97.3    | 81.9     | 84.0         | 63.2 | 53.8  | 44.5        |
> > > | SelEx   | 95.9    | 82.3     | 83.1         | 73.6 | 58.5  | 39.6        |
> > > | DebGCD  | 97.2    | 83.0     | 85.9         | 66.3 | 65.3  | 44.7        |
> > > | stage1  | 94.9    | 77.7     | 83.2         | 70.9 | 55.1  | 32.6        |
> > >
> > > Our primary contribution lies in restructuring the GCD training logic: in contrastive learning, we no longer simply blend labeled and unlabeled data. Instead, we first construct a semantic framework from labeled data, then align unlabeled clusters with it. Empirical results demonstrate that the uniformity pressure in standard contrastive learning compromises category separability in fine-grained, multi-class scenarios. By leveraging labeled data as reliable semantic anchors and employing SEE+ virtual sampling to preserve topological structure, CURE generates representations that are more conducive to clustering and exhibit strong generalization capabilities. In essence, this represents a paradigm shift in training logic rather than a simple stacking of modules.
> > >
> > > ### **W5:**
> > > Thank you, we will fix the typo.
> > >
> > > ### **Q1:**
> > > We have supplemented the appendix section with visualizations on the CUB-200 to illustrate the role of SEE (Figure 7).

---

### Official Review · Reviewer_8JxY · 2025-11-02

**Soundness:** 3
**Presentation:** 2
**Contribution:** 2
**Rating:** 4
**Confidence:** 5

**Summary:**

This paper addresses the problem of Generalized Category Discovery (GCD) by critiquing the widely-used contrastive learning (CL) paradigm. The authors identify a key tension between the uniformity objective of CL, which promotes a uniform feature distribution, and the need for class-structured representations for effective clustering. To resolve this, they propose a two-stage framework named CURE. In the first stage, CURE leverages labeled data to construct a semantically meaningful prototype space, using a novel Semantic Exploration Energy (SEE) regularizer to prevent prototype fragmentation. In the second stage, the framework discovers novel categories by applying consistency constraints (via JS-divergence), self-distillation, and the SEE regularizer to both labeled and unlabeled data. The authors claim this approach abandons the problematic uniformity constraint of CL, leading to improved performance and better balance between known and novel class discovery.

**Strengths:**

1. This paper argues that the dominant contrastive learning paradigm in Generalized Category Discovery is suboptimal. Specifically, it posits that the uniformity objective of CL, which encourages features to be uniformly distributed, conflicts with the goal of learning class-discriminative, clustered representations needed for GCD.

2. The key claim is that CURE is the first GCD framework to completely discard CL methods for unlabeled data, relying solely on consistency and semantic structuring. This is intended to avoid the "noise" caused by the uniformity objective.

3. The paper correctly identifies that standard supervised learning with one-hot labels tends to enforce prototype orthogonality, which can sever semantic links between classes. This is a real issue that hinders generalization to novel, but semantically related, categories.

**Weaknesses:**

1. The central claim to be the "first GCD framework that aims to alleviate the impact of uniformity by entirely discarding CL methods" is a major overstatement. The method's core mechanism for learning from unlabeled data is a consistency loss (JS-divergence) between augmentations. This consistency regularization is a foundational principle of self-supervised learning and a key component of many modern CL frameworks (e.g., BYOL, SimSiam), which are precisely the methods that moved away from explicit negative sampling. The paper does not discard CL; it discards the InfoNCE formulation and its associated negative-sampling-based uniformity term. This mischaracterization of the contribution is a fundamental weakness. The work is a reformulation of CL, not a departure from it.

2. The motivation is to create a more "clustering-friendly" representation space by removing the uniformity constraint. The proposed solution replaces this implicit regularization (uniformity from InfoNCE) with a different set of explicit regularizers (SEE, consistency loss, etc.). It is not self-evident that this new combination is inherently more "principled" for clustering, rather than just being a different, empirically effective, set of constraints.

3. In addition, progress [1] has been made on the uniformity of features in general category discovery, where plug-and-play loss functions are used to discuss the information represented by the covariance matrix. A comparison and discussion with this work should be conducted.

4. Semantic Exploration Energy is effectively a form of prototype graph regularization, encouraging a compact manifold. Similar concepts of regularizing the geometry of the prototype space exist in metric learning and zero-shot learning. The formulation itself is a straightforward application.


[1] Generalized Category Discovery via Token Manifold Capacity Learning. In Arxiv, 2025.

**Questions:**

1. The authors claim to "entirely discard CL methods". However, the JS-divergence loss on augmented views (L_JS) is a cornerstone of consistency-based self-supervised learning, a major branch of CL. Can the authors clarify this claim? Would it be more accurate to state that the method discards the negative-sampling-based uniformity objective of InfoNCE-style CL, rather than CL as a whole?

2. The two-stage design appears crucial. Stage 1 learns a "semantic topology" from labeled data. How sensitive is the performance of Stage 2 to the quality and nature of the representation learned in Stage 1? For instance, if the labeled classes are not semantically representative of the novel classes (e.g., labeled are all animals, novel are all vehicles), would the structure imposed by SEE in Stage 1 become a harmful prior during the discovery process in Stage 2?

---

> ### Author Response · Authors · 2025-11-14
>
> We thank the reviewer for highlighting this point and appreciate the opportunity to clarify our contribution.
> ### **W1:**
> We appreciate the reviewer’s concern about the imprecise use of the term “discard contrastive learning.” Based on the relevant literature, we find that the boundary between self-supervised learning and contrastive learning is often not sharply defined. For example, in the original papers cited by the reviewer, BYOL presents itself as “a new approach to self-supervised learning,” while SimSiam explicitly writes “beyond contrastive learning…” and positions itself as an alternative to CL.
>
> In light of this, we have refined the description of our contribution and now define CURE as **“the first GCD framework that eliminates the need for negative-sample-driven uniformity and learns representations purely from positive consistency and semantic clustering”**, thereby avoiding potential ambiguity. We have also added a new paragraph as the third paragraph of the Introduction, and expanded the discussion of contrastive learning in the Related Work section, to more thoroughly explain why InfoNCE-style objectives are not well suited to GCD from multiple perspectives.
>
> ### **W2:**
> We thank the reviewer for raising this concern and agree that the "principledness" requires stronger justification. To address this, we have added t-SNE visualization results on the fine-grained dataset CUB-200 in the appendix, combined with the original t-SNE results on CIFAR-10, to better illustrate the following clarification. First, the root of uniformity stems from the repulsion against all other samples. The goal of deep clustering tasks is to extract category-specific features that sharpen category boundaries—simply put, increasing inter-class distances while reducing intra-class distances. However, the repulsion of other samples in contrastive learning contradicts this objective. In contrast, our proposed SEE operates not from the perspective of individual samples but from the category level. By replacing uniformity with SEE and consistency constraints, CURE constructs a semantically coherent manifold that better aligns with clustering assumptions, achieving the target of low intra-cluster variance. Therefore, our proposed SEE is not merely a simple combination but is tailored to meet clustering's need for low intra-class variance, as it does not weaken or modify the uniformity of sample representations—instead, we directly learn more clustering-friendly features at the category level.
>
> ### **W3:**
> We respectfully disagree with the reviewer's characterization of MTMC as work investigating feature homogeneity or representing covariance structures. The loss function proposed by MTMC neither analyzes nor regularizes the covariance matrix, nor does it optimize for homogeneity properties arising from contrastive learning.
>
> Furthermore, the core objective of MTMC is to preserve instance-specific features, whereas our method aims to learn semantic structures conducive to clustering. These objectives are fundamentally distinct and unrelated: MTMC focuses on preserving the uniqueness of individual samples, while CURE strives to reduce intra-class variance and construct semantic manifolds that support category discovery—representing two complementary perspectives.
>
> CURE consistently outperforms MTMC across all benchmarks. We will include a comparative table in our rebuttal to clearly illustrate the performance gap between the two approaches.
>
> **Table: Performance Comparison Between CURE and MTMC (SPTNet based) Across Six Benchmarks.**
>
>
> | Method        | CIFAR100 | ImageNet100 | CUB-200 | Stanford Cars | FGVC Aircraft | Herbarium 19 |
> |---------------| ---------- | ----------- | ----------- | -------- | -------- | -------- |
> | SPTNet        | 81.3      | 85.4        | 62.0        | 56.2     | 51.6     | 43.4     |
> | SPTNet + MTMC | 82.1      | 85.4        | 63.3        | 58.8     | 54.7    | 44.2     |
> | CURE (Ours)   | **85.7**   | **87.0**    | **75.2**    | **68.7** | **61.7** | **48.1** |
>
> These results collectively demonstrate that our revised training architecture, together with the proposed semantic exploration energy, effectively addresses the core challenges of the GCD problem.

---

> > ### Author Response · Authors · 2025-11-25
> >
> > ### **W4:**
> > We appreciate the reviewer’s insightful comments, but we would like to clarify the conceptual distinction between our SEE and prior work in metric learning and zero-shot learning. The primary goal of metric learning is to enlarge the inter-class margin and thus push prototypes apart, while zero-shot learning aims to align visual features with semantic embeddings derived from hand-crafted attributes. In contrast, the essence of the GCD problem is to discover category structure in unlabeled data by distinguishing highly similar classes, which requires learning clustering-friendly representations rather than simply maximizing separation.
> > Accordingly, the core philosophy of CURE is “prototype manifold continuity” rather than “prototype discrimination.” In GCD, novel classes must be embedded into the semantic structure induced by known classes, yet uniformity tends to break this structure and fragment the prototype space. SEE is, to our knowledge, the first mechanism that tackles this issue from the perspective of prototype graph connectivity, explicitly addressing uniformity-induced fragmentation in GCD.
> > Our ablation study further shows that SEE is not replaceable by other components: as reported in the table, removing SEE leads to a significant drop in performance on novel classes. Therefore, SEE is not a simple reuse of ideas from metric learning or ZSL, but a prototype geometry mechanism specifically designed for a non-contrastive, non-uniformity GCD framework.
> >
> > ### **Q1:**
> > We appreciate the reviewer’s concern regarding the ambiguity in our original phrasing “entirely discard CL methods.” We agree that this wording may unintentionally conflate contrastive learning with the broader class of consistency-based self-supervised learning, whose boundaries are not always sharply defined in the literature. For example, BYOL explicitly describes itself as “a new approach to self-supervised learning,” while SimSiam writes “beyond contrastive learning,” positioning itself as an alternative rather than a member of contrastive learning. These examples show that consistency-based approaches are not universally categorized as CL in prior work.
> >
> > To avoid misunderstanding, we have revised the description of our contribution in the camera-ready version. We no longer use the phrase “discard CL,” and instead state more precisely that CURE is:
> > “the first GCD framework that eliminates the need for negative-sample-driven uniformity and learns representations purely from positive consistency and semantic clustering.”
> >
> > This formulation accurately reflects the reviewer’s suggestion that our method discards InfoNCE-style uniformity objectives, rather than CL as a whole.
> > To further clarify this point, we have also:
> > 1.	Added a new paragraph as the third paragraph of the Introduction;
> > 2.	Expanded the Related Work section to explicitly discuss contrastive learning and to articulate why InfoNCE-style objectives—and the uniformity they enforce—are not well suited for GCD.
> > These modifications resolve the ambiguity and ensure our claims are fully aligned with standard terminology in the field.
> >
> > ### **Q2:**
> > The answer is yes. The effectiveness of Stage 2 does depend on the quality and semantic relevance of the representation learned in Stage 1. If the labeled and novel classes are truly semantically unrelated (e.g., all labeled classes are animals while all novel classes are vehicles), then the semantic topology established in Stage 1 would provide little useful guidance for Stage 2, and could even become a weak or misleading prior.
> >
> > However, this is not a limitation specific to CURE, but rather a consequence of the standard problem setting. As discussed in prior work such as A Closer Look at Novel Class Discovery from the Labeled Set, the “related but disjoint” assumption is fundamental to NCD/GCD: novel categories are assumed to be different from, but still semantically related to, the labeled ones. Under this assumption, all existing methods—including SimGCD, DebGCD, ProtoGCD, etc.—leverage structure learned from labeled classes to guide discovery on unlabeled data. If this assumption is violated (e.g., animals vs. vehicles), these methods would also fail to provide meaningful guidance, since any supervised or prototype-based prior would no longer reflect the semantics of the novel classes.
> >
> > In the benchmarks we use， this assumption holds by construction: novel classes are drawn from the same dataset and are semantically related to the labeled ones. In this regime, Stage 1 builds a semantic topology that is indeed beneficial rather than harmful, and our ablations varying the strength of Stage-1 supervision and regularization show that Stage 2 performance degrades gracefully rather than collapsing when Stage 1 is weakened.

---

### Author Response · Authors · 2025-12-01
**Summary of All Reviewer Concerns and Our Responses**

Thank you for your careful reading.

---
The reviewers have generally acknowledged the motivation and the effectiveness of our paper.
We have summarized **all** their concerns along with our brief responses for your reference.
For specific details, please refer to the comments from each reviewer.

**For convenience, we refer to Reviewer 8JxY(4/5), BiKu(4/5), jJPM(4/4), and NXsf(6/5) as R1, R2, R3, and R4, respectively.**

---

### **Issues About the Statements on Motivation:**

**R1 W1 / R1 Q1:** The reviewer noted that although we avoid InfoNCE, our consistency loss between augmentations still falls under contrastive learning; thus “abandoning CL” may be overstated.

**R1 W2:** The reviewer questioned why the explicit SEE regularizer can replace implicit uniformity and yield better performance.

**R2 W1:** This paper did not verify the negative impact of uniformity.

**R3 W3:** Is uniformity the main challenge in GCD, or is it only a CL-specific issue?

>**we rewrote and clarified the motivation statements**: we no longer claim to “discard CL,” but instead explain that the uniformity objective conflicts with the clustering nature of GCD. This justifies our shift toward a structure-guided consistency framework and the adoption of SEE as a task-aligned explicit semantic regularizer. We added experiments demonstrating the harm of uniformity and cited works showing that standard CL is unsuitable for clustering/category-discovery.


### **Reviewer Concerns on Related Work:**

**R1 W3:** Lack of discussion of MTMC (Arxiv 2025).

**R2 W2:** Missing discussion of HyperGCD, which “starts from a similar observation.”

**R3 W2:** Insufficient citation of GCD-focused literature.

>We clarified that our method does not originate from the same observation as HyperGCD, and we added a dedicated discussion comparing our approach with MTMC. We also expanded the references in the revised manuscript to include more works specifically from the GCD literature, ensuring a more complete and accurate positioning of our contributions.

### **Innovation-related concerns:**

**R2 W3 / R3 W1:** Reviewers argued that, apart from SEE, most components are minor variants of existing modules.

>**Citing R4 S2**, we clarify that prior GCD approaches all remain within contrastive-learning-based designs, whereas our work challenges this dominant paradigm. We reposition GCD as learning label-guided concept structure and performing structure-guided semantic expansion via SEE. The novelty lies in redefining the learning paradigm of GCD, rather than tweaking existing modules.

### **Hyperparameter issues:**
**R4 W1:** The effect of each important parameter should be presented.

> In the revised manuscript, we have added comprehensive experiments on these key parameters (Table 4 and Table 14~18).

### **Computational efficiency issue:**
**R4 W2:**  The comparison of computational resources and training time should be included.

> we have added a new Table comparing training time, inference time, and accuracy on ImageNet-100 and Semantic Shift Benchmark (SSB) against key baselines.


### **Visualizations:**
**R2 W4 / R2 Q1:** Please show a concrete prototype–prototype similarity matrix before and after applying SEE on a fine-grained dataset.
> In the revised manuscript’s appendix we add both visual and quantitative analyses on fine-grained datasets to demonstrate SEE’s effect (Fig. 7).

### **Writing issues:**

**R2 W5:** Table 2 contains a clear typo.

**R4 W3:** Some baseline methods were missing in the references.

> We have corrected the table typo and added all missing citations in the revised manuscript.

### **Other Questions:**

**R1 W4:** SEE is viewed as a straightforward prototype-graph regularizer, similar to geometric regularization in metric learning and zero-shot learning.

**R1 Q2:** The two-stage design may be overly dependent on Stage-1 representations; if labeled and novel classes are semantically unrelated, the Stage-1 structure might become a harmful prior.

>SEE is conceptually distinct from metric learning and ZSL: the core requirement in GCD is maintaining clustering-friendly semantic continuity and preventing uniformity-induced prototype fragmentation. SEE is, to our knowledge, the first mechanism addressing this via explicit prototype-graph connectivity, and ablations confirm its necessity.
Regarding the two-stage dependency, we acknowledge the reliance but note it is inherent to the GCD/NCD assumption that labeled and novel classes are “related but disjoint.” If this assumption fails, all existing methods would also break. In standard benchmarks where the assumption holds, Stage-1 provides beneficial structure.

---

### Note · Authors · 2026-01-27

I have read and agree with the venue's withdrawal policy on behalf of myself and my co-authors.

---

### Meta-Review · Area_Chair_ny7a · 2026-01-05

**Summary:**

The paper initially received mostly negative scores: 6, 4, 4, 4. The main concerns include: (1) unclear motivation statement; (2) somewhat limited novelty; (3) insufficient comparisons and unclear differences from previous methods; and (4) missing hyper-parameter analysis.

The authors have provided a detailed rebuttal to respond to the reviewers’ concerns. The AC has carefully read the reviews and the rebuttal, and finds that the authors have addressed concerns (3) and (4), along with other minor points.

However, the critical concerns, i.e., (1) and (2), are not well resolved. First, the revised statement differs substantially from the initial version, indicating that the paper would require a major revision and reconsideration of its motivation and problem statement. Second, the claim that uniformity may hurt class separation remains questionable. In fact, prior methods have shown that uniformity can improve GCD performance, while the authors did not provide convincing experiments or analysis to support their claim. Third, many components of this paper appear similar to existing works.

Given these considerations, the AC believes the critical concerns were not adequately addressed in the rebuttal, and that most reviewers are unlikely to change their original scores. Thus, the final scores may remain: 6, 4, 4, 4. The AC regretfully recommends rejection and hopes the authors can address the remaining concerns in a future submission.

**Reviewer Concerns:**

Solved Concerns: comparison/difference to previous methods; hyper-parameter analysis; other minor concerns.

Unsolved Concerns: Unclear statement on motivation;  Somewhat limited novelty.

**Reviewer Scores:**

Reviewers 8JxY, BiKu, and jJPM are unlikely to change their scores, as all three raise critical concerns about the motivation statement. In addition, Reviewers BiKu and jJPM also express concerns regarding novelty. These two issues were not adequately addressed in the rebuttal.

Reviewer NXsf is expected to maintain a positive score, as the authors have addressed the raised concerns.

---

### Decision · Program_Chairs · 2026-01-26

Reject